# LLaDA 1.5: Variance-Reduced Preference Optimization for Large Language Diffusion Models

## Abstract

Diffusion large language models present a promising paradigm to language modeling, yet their alignment remains underexplored, particularly in systematic theoretical analysis and comprehensive empirical validation on general tasks. In this paper, we identify a primary challenge for this problem: the high variance in Evidence Lower Bound (ELBO)-based likelihood estimates required for preference optimization. To address this issue, based on Direct Preference Optimization (DPO), we propose *Variance-Reduced Preference Optimization* (VRPO), a framework that formally analyzes the bias and variance of the preference optimization loss and gradient, showing both are governed by a score-estimator variance. Building on this foundation, we introduce multiple unbiased variance reduction strategies, including optimal budget allocation and antithetic sampling, to improve the alignment performance. We demonstrate the effectiveness of VRPO by applying it to LLaDA, a large-scale diffusion language model. The resulting model, LLaDA 1.5, outperforms its SFT-only predecessor consistently across mathematical (GSM8K +4.7), code (HumanEval +3.0, MBPP +1.8), and alignment (IFEval +4.0, Arena-Hard +4.3) benchmarks. Furthermore, LLaDA 1.5 demonstrates a highly competitive mathematical performance compared to other strong language MDMs and ARMs.

## 1 Introduction

Recently, masked diffusion models (MDMs) (Sohl-Dickstein et al., 2015; Austin et al., 2021a; Campbell et al., 2022; Meng et al., 2022; Lou et al., 2023; Sahoo et al., 2024; Shi et al., 2024; Ou et al., 2024) have achieved significant progress in language modeling. By optimizing the evidence lower bound (ELBO) or its simplified variants, MDMs have demonstrated comparable or even superior performance to autoregressive models (ARMs) at a small scale (Lou et al., 2023; Ou et al., 2024; Nie et al., 2024). Explorations on the scaling properties have also revealed MDMs' excellent scalability in various downstream tasks (Nie et al., 2024; Gong et al., 2024; Nie et al., 2025), achieving competitive results to representative ARMs of the same size (e.g., LLaMA 3 (Dubey et al., 2024)).

Motivated by the success of aligning ARMs with human preferences (Schulman et al., 2017; Ziegler et al., 2019; Ouyang et al., 2022; Rafailov et al., 2023; Shao et al., 2024; Guo et al., 2025), recent work has begun to explore MDM alignment (Zekri and Boullé, 2025; Borso et al., 2025; Zhao et al., 2025; Huang et al., 2025; Yang et al., 2025; Gong et al., 2025; Tang et al., 2025). Notably, most current methods adapt existing alignment frameworks to MDMs, introducing various likelihood approximation methods without providing pertinent theoretical analysis. Moreover, they primarily focus on specialized tasks such as reasoning and code generation–which, while important–leave broader alignment tasks underexplored that are essential for future diffusion language model development.

In this paper, we systematically study the challenge of aligning MDMs based on direct preference optimization (DPO) (Rafailov et al., 2023), for its simplicity and notable empirical performance. The key challenge is that the original DPO formulation requires exact log-likelihoods, which are intractable for diffusion models. A natural solution under this scenario is to approximate these log-likelihoods with their evidence lower bounds (ELBOs), which introduce nested expectations over diffusion time and masked data. This substitution yields an ELBO-based preference score expressed as a linear combination of four ELBO terms (see Eq.(7)).

In practice, these ELBO terms are estimated via a doubly Monte Carlo method (Titsias and Lázaro-Gredilla, 2014; Dai et al., 2014). We demonstrate that this estimation introduces additional bias and

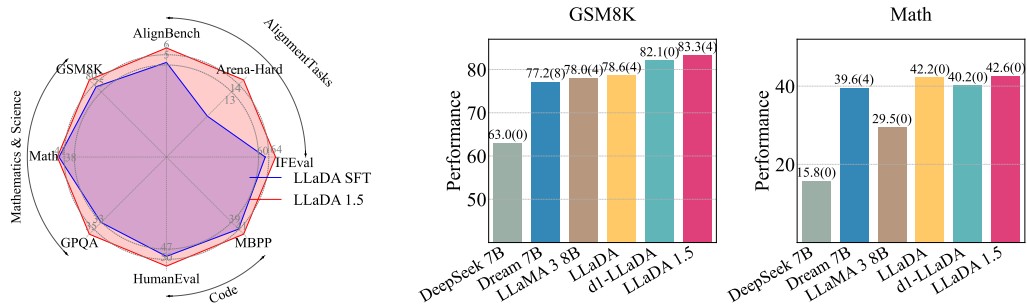

Figure 1: **Benchmark results.** The left panel shows that LLaDA 1.5 improves LLaDA consistently and significantly on various benchmarks. The right panel demonstrates that LLaDA 1.5 has a highly competitive mathematical performance compared to strong language MDMs and ARMs.

variance into the preference optimization loss and gradient. To mitigate these errors, our theoretical analysis reveals a crucial insight: the introduced bias and variance are governed by the variance of the preference score estimator. This finding underscores the need to control this variance for stable and effective preference optimization.

Building upon this, we introduce *Variance-Reduced Policy Optimization* (VRPO), a method integrating principled techniques to reduce the variance of the preference score estimator: **(1)** increasing the sampling budget for ELBOs, **(2)** allocating the sampling budget across distinct diffusion timesteps with one masked sample per timestep, and **(3)** applying antithetic sampling (Kroese et al., 2013) between ELBO estimates of the model and reference policies. These techniques have been theoretically proven to reduce the variance of the score estimator in an unbiased manner and empirically validated in both synthetic (as in Figure 2) and large-scale real-world ablation studies (as in Section 4.2). We further discuss on potential generalization of our variance reduction techniques to other alignment algorithms such as PPO and GRPO (Schulman et al., 2017; Shao et al., 2024).

Finally, we show the effectiveness of VRPO by applying it to LLaDA 8B Instruct (Nie et al., 2025), a leading language MDM, using 350k preference pairs. As shown in Figure 1, the resulting model, LLaDA 1.5, improves LLaDA consistently on mathematics, coding, and alignment tasks. In addition, LLaDA 1.5 maintains a highly competitive mathematical performance compared to other strong MDMs (Nie et al., 2025; Ye et al., 2025; Zhao et al., 2025) and ARMs (Dubey et al., 2024; Bi et al., 2024), achieving the highest score on Math. These results demonstrate the effectiveness of our variance reduction method and establish a foundation for further development of language MDMs.

## 2 PRELIMINARIES

### 2.1 ALIGNMENT METHODS

Traditional alignment approaches (Ziegler et al., 2019; Ouyang et al., 2022) consist of two stages.

**Reward modeling.** In the first stage, a static dataset of preference comparisons $\mathcal{D} = \{(x, y_w, y_l)\}$ is constructed. For each prompt $x$, $y_w$ denotes the human-preferred response and $y_l$ denotes the less preferred one, respectively. A parameterized reward model $r_\phi$ is trained to reflect these preferences by minimizing the following objective based on Bradley-Terry formulation (Bradley and Terry, 1952):

$$\mathcal{L}_{\text{Reward}}(\phi) \triangleq -\mathbb{E}_{(x,y_w,y_l)\sim\mathcal{D}} \left[ \log \sigma \left( r_\phi(x, y_w) - r_\phi(x, y_l) \right) \right], \qquad (1)$$

where $\sigma(\cdot)$ is the sigmoid function. This encourages $r_\phi$ to assign higher scores to preferred responses.

**Reinforcement Learning (RL).** In the second stage, the language model policy $\pi_\theta(y \mid x)$, which defines the probability of generating response $y$ given prompt $x$, is then optimized via RL to maximize:

$$\max_{\pi_\theta} \mathbb{E}_{x\sim\mathcal{D},\, y\sim\pi_\theta(\cdot|x)} \left[ r_\phi(x, y) \right] - \beta \, \mathbb{D}_{\text{KL}} \left( \pi_\theta(\cdot \mid x) \,\|\, \pi_{\text{ref}}(\cdot \mid x) \right), \qquad (2)$$

where $\pi_{\text{ref}}$ is a fixed reference policy, often chosen as a frozen SFT model, and $\beta$ is a coefficient controlling the regularization strength. Notably, in autoregressive models (ARMs), both sampling and likelihood evaluation for the policy are exactly characterized by the language model distribution.

**Direct Preference Optimization (DPO).** DPO (Rafailov et al., 2023) offers a simplified alternative to the two-stage paradigm above by avoiding explicit reward model training, while maintaining both theoretical grounding and strong empirical performance (Grattafiori et al., 2024). The DPO objective is to minimize $\mathcal{L}_{\mathrm{DPO}}(\theta) \triangleq \mathbb{E}_{(x,y_w,y_l)\sim\mathcal{D}} \left[ \ell_{\mathrm{DPO}}(x, y_w, y_l; \theta) \right]$, where

$$\ell_{\mathrm{DPO}}(x, y_w, y_l; \theta) \triangleq -\log \sigma \left( \beta \log \frac{\pi_\theta(y_w \mid x)}{\pi_{\mathrm{ref}}(y_w \mid x)} - \beta \log \frac{\pi_\theta(y_l \mid x)}{\pi_{\mathrm{ref}}(y_l \mid x)} \right). \tag{3}$$

## 2.2 Masked Diffusion Models

Masked Diffusion Models (MDMs) define a model distribution via a forward–reverse framework (Sohl-Dickstein et al., 2015; Austin et al., 2021a). Starting from the original input at $t = 0$, the forward process progressively masks the input tokens with a masking probability increasing over time, producing a fully masked sequence at $t = 1$. The reverse process learns to denoise this sequence by iteratively predicting the masked tokens as time reverses from $t = 1$ to $t = 0$. This framework enables principled modeling of complex data distributions, offering a feasible exploration for non-autoregressive generation approaches.

**Likelihood estimation in MDMs.** Unlike ARMs, the exact log-likelihood $\log \pi(y \mid x)$ in MDMs is often approximated by its evidence lower bound (ELBO) (Lou et al., 2023; Ou et al., 2024; Shi et al., 2024; Sahoo et al., 2024) as follows:

$$\mathcal{B}_\pi(y \mid x) \triangleq \mathbb{E}_{t\sim\mathcal{U}[0,1]}\mathbb{E}_{y_t\sim q(y_t|t,y,x)} \left[ \ell_\pi(y_t, t, y \mid x) \right] \leq \log \pi(y \mid x), \tag{4}$$

where $q(y_t \mid t, y, x)$ denotes the forward diffusion process at time $t$ given the full response $y$ and prompt $x$, and $\ell_\pi$ represents the per-step loss of the mask prediction model, which admits multiple equivalent formulations elaborated in Appendix B. Notably, for a continuous-time diffusion process (or equivalently, when the step size is infinitesimal), the *bias* of the ELBO for a well-trained model relative to the exact likelihood is negligible, as demonstrated in prior works (Sohl-Dickstein et al., 2015; Ho et al., 2020; Song et al., 2020). From now on, we omit the conditions on $x$ for brevity.

Computing $\mathcal{B}_\pi(y)$ exactly is generally intractable due to the double expectations. In practice, we approximate it by a *doubly Monte Carlo* method. Letting $n_t$ and $n_{y_t}$ be the numbers of samples for timesteps and masked data per timestep, we draw:

$$\mathcal{S}_t \triangleq \{t^{(j)}\}_{j=1}^{n_t} \overset{\text{i.i.d.}}{\sim} \mathcal{U}[0,1] \quad \text{and} \quad \mathcal{S}_{y_{t(j)}|y} \triangleq \{y_{t^{(j)}}^{(k)}\}_{k=1}^{n_{y_t}} \overset{\text{i.i.d.}}{\sim} q(y_t \mid t^{(j)}, y), \; j = 1, \ldots, n_t, \tag{5}$$

where the masked data for different timesteps are independently sampled, i.e., given $y$ and $\mathcal{S}_t$, $\mathcal{S}_{y_{t(j)}|y} \perp\!\!\!\perp \mathcal{S}_{y_{t(j')}|y}$ for any $j \neq j'$. The ELBO is then estimated by:

$$\widehat{\mathcal{B}}_\pi(y) \triangleq \frac{1}{n_t} \sum_{j=1}^{n_t} \frac{1}{n_{y_t}} \sum_{k=1}^{n_{y_t}} \ell_\pi(y_{t^{(j)}}^{(k)}, t^{(j)}, y), \tag{6}$$

which is an average of mask-prediction loss computed over a total of $n = n_t \times n_{y_t}$ masked data. The estimator in Eq. (6) is an unbiased approximation for the ELBO following from the linearity of expectations. However, due to computational constraints, large values for $n$ are typically not used. As a result, the variance of the estimator must be considered. Particularly, in the context of MDMs' DPO, this presents unique challenges for optimization, as will be discussed in the next section. In this work, we explore how to mitigate the negative effects of this ELBO estimation variance on preference optimization, considering both scenarios with scalable and fixed computational budgets.

**Large-scale language MDMs.** LLaDA (Nie et al., 2025) is an 8B-parameter masked diffusion model for language generation. LLaDA is pretrained on 2.3 trillion tokens and fine-tuned on 4.5 million pairs of SFT data. It exhibits outstanding capabilities comparable with representative ARMs (Dubey et al., 2024) in scalability, in-context learning, and instruction-following. In this paper, we adopt LLaDA 8B as a base model to explore and validate alignment methods for MDMs.

## 3 Method

We investigate how to align MDMs with human preferences using the DPO framework (Rafailov et al., 2023). To address the intractability of the required log-likelihoods, we approximate them by

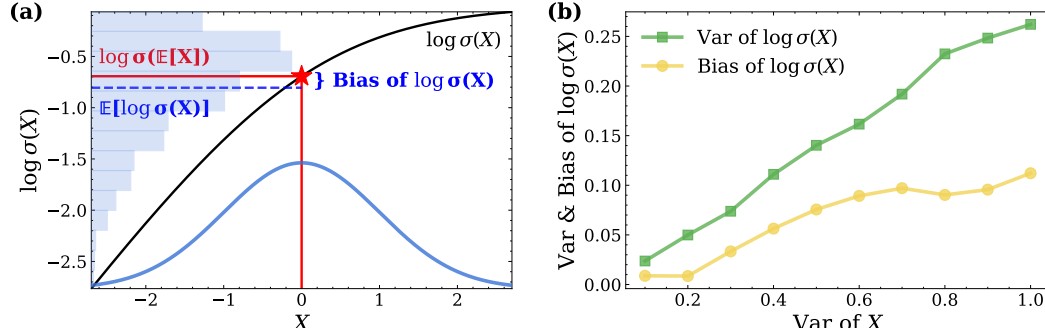

Figure 2: **Toy example.** (a) Although $X$ is an unbiased estimator for $\mathbb{E}[X]$, $\log \sigma(X)$ is not an unbiased estimator for $\log \sigma(\mathbb{E}[X])$. Non-linear transformation introduces a gap between $\mathbb{E}[\log \sigma(X)]$ and $\log \sigma(\mathbb{E}[X])$ (blue and red horizontal lines). (b) Both the bias and variance of $\log \sigma(X)$ exhibit monotonic trends with $\mathbb{V}[X]$, supporting the insight to jointly reduce these errors by reducing $\mathbb{V}[X]$.

ELBO estimators. We prove that the bias and variance of the resulting loss and its gradient can be bounded by the variance of a score estimator (a linear combination of four ELBOs). Based on this, we propose *Variance-Reduced Preference Optimization* (VRPO), integrating multiple unbiased variance reduction techniques for better alignment. We also discuss potential extension beyond DPO.

### 3.1 SUBSTITUTING LIKELIHOODS WITH ELBOS IN DPO

Let us begin by adapting the DPO loss in Eq. (3) by substituting log-likelihoods with their ELBOs:

$$\ell_{\text{DPO}-\text{E}}(y_w, y_l; \theta) \triangleq -\log \sigma \Big( \beta \left( \mathcal{B}_{\pi_\theta}(y_w) - \mathcal{B}_{\pi_{\text{ref}}}(y_w) \right) - \beta \left( \mathcal{B}_{\pi_\theta}(y_l) - \mathcal{B}_{\pi_{\text{ref}}}(y_l) \right) \Big). \quad (7)$$

We refer to the term in red as the ELBO-based preference *score* and denote it by $s_\theta(y_w, y_l)$.

Intuitively, the loss encourages the current model $\pi_\theta$ to better prefer $y_w$ over $y_l$ than reference $\pi_{\text{ref}}$ by comparing the ELBOs. As discussed around Eq. (4), the ELBO provides a principled approximation to the log-likelihood with negligible bias. Moreover, the structure of the DPO loss—specifically its symmetric form and the smoothness of the sigmoid function—inherently helps mitigate the overall approximation gap, making $\ell_{\text{DPO}-\text{E}}$ a reliable surrogate for the original DPO objective.

In practice, each ELBO in Eq. (7) is estimated by Eq. (6). The resulting estimated loss is:

$$\widehat{\ell}_{\text{DPO}-\text{E}}(y_w, y_l; \theta) \triangleq -\log \sigma \Big( \beta \left( \widehat{\mathcal{B}}_{\pi_\theta}(y_w) - \widehat{\mathcal{B}}_{\pi_{\text{ref}}}(y_w) \right) - \beta \left( \widehat{\mathcal{B}}_{\pi_\theta}(y_l) - \widehat{\mathcal{B}}_{\pi_{\text{ref}}}(y_l) \right) \Big), \quad (8)$$

where we denote the score estimator, highlighted in red, by $\hat{s}_\theta(y_w, y_l)$, and we use $\mathcal{S}_{\hat{s}|y_w, y_l}$ to denote the stochastic sampling involved in this estimation.

Notably, for a fixed pair of preference data $y_w, y_l$, the stochastic sampling in this score estimator introduces randomness into the estimated loss, making it a random variable over $\mathcal{S}_{\hat{s}|y_w, y_l}$, and thereby introduces *variance* into both the loss and its gradient. Besides, due to the nonlinearity of $\log \sigma(\cdot)$, this also results in additional *bias* between $\mathbb{E}[\log \sigma(\hat{s}_\theta(y_w, y_l))]$ and the target $\log \sigma(s_\theta(y_w, y_l)) = \log \sigma(\mathbb{E}[\hat{s}_\theta(y_w, y_l)])$ (see Figure 2 (a) for an intuitive illustration), although $\hat{s}_\theta$ itself is an unbiased estimator for the true score $s_\theta$ (formally explained in Appendix C.2.1).

In the remainder of this section, we address these two problems by first establishing how the variance of the score estimator governs the introduced bias and variance, and then proposing multiple principled variance reduction strategies to mitigate them. For clarity, we focus on the loss analysis in the main paper and defer the analogous gradient analysis to Appendix C.4.

### 3.2 VARIANCE-REDUCED PREFERENCE OPTIMIZATION

The following theorem demonstrates how the bias and variance of the empirical loss can be directly bounded in terms of the variance of the score estimator. Intuitively, the proof (see Appendix C.2.2) utilizes the Lipschitz continuity of $\log \sigma(\cdot)$ and the unbiasedness of $\hat{s}_\theta$, which ensures that the

variability in $\hat{s}_\theta$ leads to controlled changes in the loss and keep it close to the true objective. Tightness analysis of these upper bounds is provided in Appendix C.2.3.

**Theorem 1.** *Given a pair of preference data $y_w, y_l$, the bias and variance of $\widehat{\ell}_{\mathrm{DPO-E}}(y_w, y_l; \theta)$ over stochastic sampling in the score estimation can be bounded as:*

$$\mathbb{E}_{\mathcal{S}_{\hat{s}|y_w,y_l}}\left[\left|\ell_{\mathrm{DPO-E}}(y_w, y_l; \theta) - \widehat{\ell}_{\mathrm{DPO-E}}(y_w, y_l; \theta)\right|\right] \le \sqrt{\mathbb{V}_{\mathcal{S}_{\hat{s}|y_w,y_l}}[\hat{s}_\theta(y_w, y_l)]},$$

$$\mathbb{V}_{\mathcal{S}_{\hat{s}|y_w,y_l}}\left[\widehat{\ell}_{\mathrm{DPO-E}}(y_w, y_l; \theta)\right] \le 4\mathbb{E}_{y_w,y_l}\left[\mathbb{V}_{\mathcal{S}_{\hat{s}|y_w,y_l}}[\hat{s}_\theta(y_w, y_l)]\right].$$

In the toy example shown in Figure 2 (b), we plot how the variance of a random variable $X$ influences the bias and variance of $\log \sigma(X)$. These curves exhibit trends that align well with Theorem 1.

Collectively, these findings suggest that one can simultaneously mitigate both errors by reducing the variance of $\hat{s}_\theta$. To do this, we present VRPO, illustrated in Figure 3, a set of principled techniques designed to reduce the variance of the score estimator as follows:

**(1) Sampling budget:** Increase the number of samples $n = n_t \times n_{y_t}$ used to estimate each ELBO.

**(2) Optimal allocation:** Allocate the full budget to timesteps by setting $n_t = n$ and $n_{y_t} = 1$.

**(3) Antithetic sampling:** Share the same sampled timesteps and masked data between the ELBO estimates of the current policy $\pi_\theta$ and the reference policy $\pi_{\mathrm{ref}}$ for the same input $y_w$ or $y_l$.

Practically, the first component *increases the FLOPs* of preference optimization by a factor of $n$, while the latter two components *incur no additional computational cost*: optimal allocation redistributes the existing samples across timesteps without increasing the total sample count, and antithetic sampling reuses samples across ELBO estimates, effectively serving as a "free lunch" for variance reduction. In our default experimental setting, where $n$ is set to be 8, the additional overhead is fully affordable relative to the overall pretraining cost as discussed in Section 4, and ablation studies under both scalable and fixed computational budgets are provided in Section 4.2.

Theoretically, all of these techniques reduce the variance of $\hat{s}_\theta$ without introducing bias. Main analysis is presented below, with proofs and unbiasedness examinations deferred to Appendix C.3.

We first observe the variance of the score estimator by unrolling it according to the definition in Eq. (8) (where subscripts of variances and square brackets $[\cdot]$ are omitted for brevity):

$$\mathbb{V}\hat{s}_\theta(y_w, y_l) =$$
$$\beta^2 \sum_{y \in \{y_w, y_l\}} \left[ \mathbb{V}\widehat{\mathcal{B}}_{\pi_\theta}(y) + \mathbb{V}\widehat{\mathcal{B}}_{\pi_{\mathrm{ref}}}(y) - 2\mathrm{Corr}\left(\widehat{\mathcal{B}}_{\pi_\theta}(y), \widehat{\mathcal{B}}_{\pi_{\mathrm{ref}}}(y)\right)\sqrt{\mathbb{V}\widehat{\mathcal{B}}_{\pi_\theta}(y)\mathbb{V}\widehat{\mathcal{B}}_{\pi_{\mathrm{ref}}}(y)} \right]. \quad (9)$$

This decomposition reveals two strategies to reduce $\mathbb{V}\hat{s}_\theta$: first, decreasing the variance of each ELBO estimation; second, increasing the correlation between the ELBO estimates for the same input $y$. The techniques proposed in VRPO operate exactly according to these two strategies, as formalized below.

**Proposition 1** (Reduce the ELBO variance). *Given a total budget of $n = n_t \times n_{y_t}$ masked samples for estimating $\widehat{\mathcal{B}}_\pi(y)$, we have: (i) $\mathbb{V}\widehat{\mathcal{B}}_\pi(y) = \Theta(\frac{1}{n})$, (ii) $\mathbb{V}\widehat{\mathcal{B}}_\pi(y)$ is minimized when $n_t = n, n_{y_t} = 1$.*

**Proposition 2** (Increase the correlation). *Given any response $y$, supposing $\mathrm{Corr}\left(\widehat{\mathcal{B}}_{\pi_\theta}(y), \widehat{\mathcal{B}}_{\pi_{\mathrm{ref}}}(y)\right) > 0$ when the Monte Carlo samples $\mathcal{S}_t$ and $\{\mathcal{S}_{y_{t(j)}|y}\}_{j=1}^{n_t}$ are shared between $\widehat{\mathcal{B}}_{\pi_\theta}(y)$ and $\widehat{\mathcal{B}}_{\pi_{\mathrm{ref}}}(y)$, we have: Sharing Monte Carlo samples yields lower $\mathbb{V}\hat{s}_\theta(y_w, y_l)$ than using independent samples.*

Proposition 1 characterizes a quantitative relationship between the variance of ELBO and the sampling budget $n$ (*first technique*), and derives the optimality of allocating the entire budget across timesteps (*second technique*). Proposition 2 is inspired by the classical antithetic variates method (Kroese et al., 2013), where shared randomness is leveraged to reduce the variance of the difference between paired estimates (*third technique*). The result and its assumption are quite natural since the current and reference policies typically share initialization and exhibit similar preferences on the same inputs. This proposition primarily highlights how to leverage their positive correlation to reduce variance.

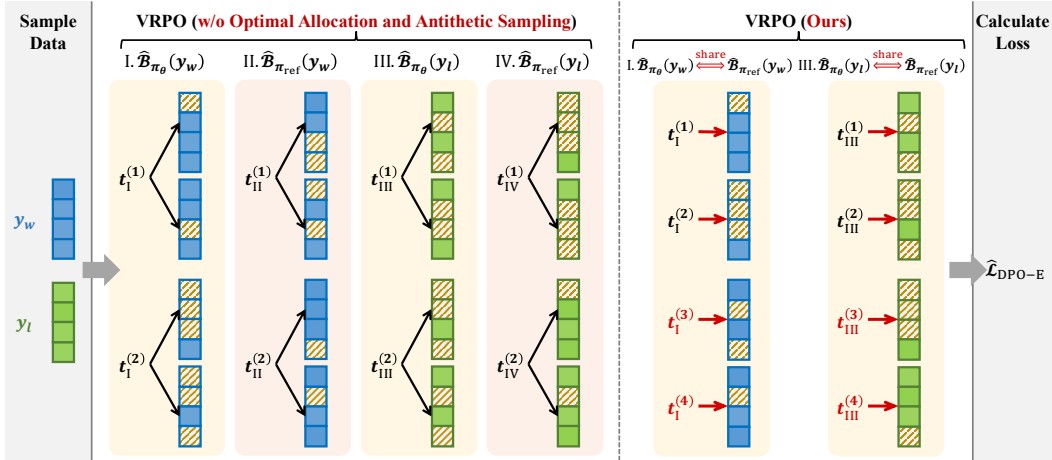

Figure 3: **Illustration of VRPO.** We compare VRPO (right) with VRPO without optimal allocation and antithetic sampling (left). VRPO allocates the sampling budget across timesteps to sample only one masked data per timestep (indicated by red arrows) and shares Monte Carlo samples between paired ELBOs (highlighted with the red annotations above the blocks).

The empirical effectiveness of VRPO is examined in Section 4. While we need to emphasize that our contribution lies not only in the proposed techniques themselves but also in the systematic analysis that motivates and supports them. Unlike approaches relying purely on empirical tuning or prior experience with continuous diffusion for visual data, our theoretical analysis provides transferable insights into variance reduction strategies, offering guidance for MDM alignment and helping rule out suboptimal implementation choices.

### 3.3 EXTENSION TO OTHER ALIGNMENT METHODS

The variance reduction techniques and analysis in VRPO are not limited to DPO, but naturally extend to other alignment algorithms that involve estimating the ELBO or subtracting two correlated ELBOs, which is a commonly encountered scenario when applying alignment to MDMs.

For example, PPO (Schulman et al., 2017) and GRPO (Shao et al., 2024) optimize variants of the objective (see Eq.(6) in Schulman et al. (2017)): $\mathbb{E}_{\pi_{\text{old}}} \left[ \frac{\pi_\theta(y|x)}{\pi_{\theta_{\text{old}}}(y|x)} \hat{A}(x, y) \right]$, where $\hat{A}(x, y)$ is the advantage function computed using a KL-penalized reward (see Eq.(2) in Ouyang et al. (2022)): $r_\theta(x, y) - \beta \log \frac{\pi_\theta(y|x)}{\pi_{\text{ref}}(y|x)}$. For both equations, when applied to MDMs, our variance reduction techniques can be directly used to reduce the variance in ELBO-based estimation for *likelihood* terms $\pi(y|x)$ or *likelihood-ratio* terms $\frac{\pi_1(y|x)}{\pi_2(y|x)}$ without introducing bias. These terms are structurally similar to those in the DPO loss (Eq. (3)), and the applicability of our techniques is supported by analogous analysis as in Propositions 1 and 2. In fact, the analysis becomes even simpler in these settings, as they do not involve the outer nonlinear $\log \sigma(\cdot)$ function that introduces additional challenge to providing theoretical guarantees as in DPO.

## 4 EXPERIMENTS

We align LLaDA (Nie et al., 2025) using VRPO for general tasks and implement extensive evaluation on common benchmarks. We briefly present the setup, with more details provided in Appendix E.

**Data.** We train LLaDA 8B Instruct (Nie et al., 2025) for one epoch on 350K preference pairs using VRPO, resulting in LLaDA 1.5. The data are collected internally at scale across rich scenarios and undergo several processing steps, including filtering out low-quality samples, removing duplicates via similarity matching, using a reward model to rank data, and replacing some responses with outputs from advanced LLMs. This process yields a high-quality and diverse dataset covering a wide range of topics such as writing, dialogue, knowledge Q&A, reasoning, mathematics, and coding.

Table 1: **Benchmark results.** We compare the performance of *LLaDA 1.5* against *LLaDA Instruct* (Nie et al., 2025) and *LLaDA with naive DPO* across various benchmarks, including mathematics, code, and alignment. The results show overall improvements for VRPO.

| | LLaDA 8B Instruct | LLaDA DPO | LLaDA 1.5 8B |
|---|---|---|---|
| Post-training | SFT | SFT + naive DPO | SFT + VRPO (**Ours**) |
| *Mathematics & Science* | | | |
| GSM8K | 78.6 | 80.7 (+2.1) | 83.3 (+4.7) |
| Math | 42.2 | 41.6 (-0.6) | 42.6 (+0.4) |
| GPQA | 33.3 | 34.3 (+1.0) | 36.9 (+3.6) |
| *Code* | | | |
| HumanEval | 49.4 | 48.2 (-1.2) | 52.4 (+3.0) |
| MBPP | 41.0 | 41.4 (+0.4) | 42.8 (+1.8) |
| *Alignment Tasks* | | | |
| IFEval | 62.2 | 62.0 (-0.2) | 66.2 (+4.0) |
| Arena-Hard | 10.0 | 11.9 (+1.9) | 14.3 (+4.3) |
| AlignBench | 5.4 | 5.8 (+0.4) | 5.9 (+0.5) |
| MTbench | 7.2 | 7.1 (-0.1) | 7.3 (+0.1) |

**Computational Cost.** We use a sampling budget $n = 8$ for VRPO by default. This results in roughly an 8 times increase in computation compared to methods without Monte Carlo estimation (e.g., ARMs or setting $n = 1$). Despite this, the overall cost remains modest—less than 0.5% of pre-training—making the added overhead practically acceptable. If considering a fixed computational budget, VRPO's optimal allocation and antithetic sampling techniques can still improve the effectiveness of preference optimization (relevant discussions are provided in ablation studies in Section 4.2).

**Metrics and evaluation.** Following common practice in open-source LLMs (Grattafiori et al., 2024; Yang et al., 2024; Liu et al., 2024), we conduct comprehensive evaluation of LLaDA 1.5 across three categories of tasks: mathematics and scientific reasoning (GSM8K (Cobbe et al., 2021), Math (Hendrycks et al., 2021), GPQA (Rein et al., 2023)), code generation (HumanEval (Chen et al., 2021), MBPP (Austin et al., 2021b)), and alignment (IFEval (Zhou et al., 2023), Arena-Hard (Li et al., 2024), AlignBench (Liu et al., 2023), MTBench (Zheng et al., 2023)). Specific to MDMs, there are three commonly used sampling strategies for inference, including diffusion sampling, diffusion semi-autoregressive sampling (Nie et al., 2025), and low-confidence remasking (Chang et al., 2022). Following common practice in MDM evaluation (Nie et al., 2025), we adopt the best sampling strategy for each task. Detailed descriptions of the sampling strategies employed and ablations across different sampling methods are provided in Appendix E.4.

## 4.1 BENCHMARK RESULTS

Table 1 presents benchmark results for three models: LLaDA Instruct; LLaDA with naive DPO ($n_t = 1$, $n_{y_t} = 1$, without antithetic sampling); and LLaDA 1.5 with VRPO, which fully incorporates variance-reduction techniques ($n_t = 8$, $n_{y_t} = 1$, with antithetic sampling). Appendix F lists instruction-following case studies comparing LLaDA 1.5 and LLaDA Instruct, as a supplement. Ablations for VRPO's components under a fixed compute budget are provided in Section 4.2.

As a result, LLaDA 1.5 consistently outperforms both baselines across all benchmarks, showing the overall effectiveness of VRPO on various tasks. Particularly, we observe that LLaDA 1.5 nonetheless exhibits strong mathematical performance. As also shown in the right panel of Figure 1, compared with similar-scale language MDMs and ARMs (Nie et al., 2025; Ye et al., 2025; Zhao et al., 2025; Dubey et al., 2024; Bi et al., 2024), it remains competitive and achieves the highest four-shot score on GSM8K and the highest zero-shot score on Math. Overall, these results demonstrate the effectiveness of VRPO, laying the groundwork for future work to further enhance MDMs' performance.

Table 2: **Ablation of VRPO variance reduction strategies.** We report estimator variances and benchmark results under different sampling configurations. As for biases, we refer to Figure 2 as an illustration since they are difficult to measure in practice. Results confirm that techniques in VRPO generally improve task performance, supporting the theoretical analysis in Section 3.

| | Base | Budget | | Allocation | | Antithetic |
|---|---|---|---|---|---|---|
| # Timesteps $n_t$ | 4 | 1 | 8 | 1 | 2 | 4 |
| # Masked samples $n_{y_t}$ | 1 | 1 | 1 | 4 | 2 | 1 |
| Antithetic sampling | ✓ | ✓ | ✓ | ✓ | ✓ | ✗ |
| *Variances* | | | | | | |
| Var of score estimator | 2.2 | 44.0 | 1.0 | 7.3 | 4.7 | 2183.7 |
| Var of loss | $3.1 \times 10^{-3}$ | $8.7 \times 10^{-2}$ | $2.6 \times 10^{-3}$ | $3.2 \times 10^{-2}$ | $7.3 \times 10^{-3}$ | 62.0 |
| Var of gradient | 2.5 | 13.0 | 1.6 | 4.7 | 2.5 | 10.6 |
| *Mathematics & Science* | | | | | | |
| GSM8K | 82.8 | 80.1 | 83.3 | 81.4 | 82.3 | 82.0 |
| Math | 42.3 | 41.7 | 42.6 | 41.9 | 42.4 | 42.4 |
| GPQA | 36.4 | 34.3 | 36.9 | 34.9 | 36.4 | 35.9 |
| *Code* | | | | | | |
| HumanEval | 51.2 | 50.6 | 52.4 | 48.2 | 48.8 | 47.0 |
| MBPP | 42.8 | 40.6 | 42.8 | 40.8 | 41.0 | 41.2 |
| *Alignment Tasks* | | | | | | |
| IFEval | 66.1 | 63.9 | 66.2 | 64.8 | 66.2 | 65.8 |
| Arena-Hard | 13.9 | 13.5 | 14.3 | 13.8 | 13.4 | 15.6 |
| AlignBench | 5.9 | 5.6 | 5.9 | 5.8 | 5.9 | 5.9 |
| MTbench | 7.4 | 7.0 | 7.3 | 7.0 | 7.2 | 7.2 |

## 4.2 ABLATION EXPERIMENTS

We conduct ablation studies to evaluate the impact of each variance reduction technique in VRPO. We vary sampling configurations in three factors corresponding to these components: **(1)** the sampling budget $n = n_t \times n_{y_t}$, **(2)** the allocation strategy between the number of timesteps and masked samples per timestep $n_t/n_{y_t}$, and **(3)** the use of antithetic sampling. We set the base configuration as $n = 4$, $n_t/n_{y_t} = 4/1$, with antithetic sampling used. For each configuration, we measure: **(i)** the variance of the score estimator $\mathbb{V}\hat{s}_\theta$, **(ii)** the additional variances of the loss and gradient, and **(iii)** benchmark results spanning mathematics, code, and alignment. Results are summarized in Table 2. To illustrate the impact of these techniques on the optimization process more concretely, we also provide the training loss dynamics for the ablation configurations in Appendix E.3. Details of the empirical variance computation are provided in Appendix E. We highlight key observations below.

**Effect of preference score estimator variance.** Lower variances of the score estimator generally lead to lower variances in both the loss and gradient, along with improved task performance. This empirical trend supports our theoretical insight in Theorem 1 to control the errors by $\mathbb{V}\hat{s}_\theta$.

**Increasing sampling budget.** Increasing the sampling budget $n$ consistently reduces estimator variance and improves task performance. For instance, increasing $n$ from 1 to 8 reduces $\mathbb{V}\hat{s}_\theta$ from 44.0 to 1.0 and improves GSM8K accuracy from 80.1 to 83.3, validating our finding in Proposition 1 **(i)**.

**Comparison under fixed sampling budget.** The first, fourth, and sixth columns show results under a fixed sampling budget, where the fourth and sixth columns disable the optimal allocation technique and antithetic sampling technique, respectively. For **optimal allocation**, it is shown to generally yield lower variance and better results than repeating multiple mask samples per timestep, supporting the analysis in Proposition 1 **(ii)**. For **antithetic sampling**, we observe that it leads to notable decreases in variance, confirming our prediction in Proposition 2. Despite this, we also observe that these sharp reductions in variance do not always translate into substantial improvements on downstream benchmarks. We believe this is understandable since the benchmark performance depends on two factors: *optimization* and *generalization*. VRPO is designed to improve *optimization* and has shown

effective (as further illustrated in Figure 5), whereas *generalization* is influenced by complex factors that are rarely feasible to control. We hypothesize that disabling antithetic sampling may expose the model to a broader diversity of data patterns, which could benefit certain downstream tasks.

To summarize, these results demonstrate a strong empirical correlation between the proposed techniques and variance reduction, and benchmark results further confirm their essential role in effective preference optimization, which aligns with the theoretical analysis in Section 3.

## 5 RELATED WORK

**Masked diffusion models.** MDMs are inspired by advances in discrete diffusion models (Sohl-Dickstein et al., 2015; Austin et al., 2021a), which introduced new forward and reverse transitions and enabled numerous variants (Campbell et al., 2022; Hoogeboom et al., 2021; He et al., 2022; Wu et al., 2023; Zheng et al., 2024). Empirically, MDMs can match ARMs in perplexity, and simplified objectives for masked diffusion have been proposed for efficient training (Lou et al., 2023; Sahoo et al., 2024; Shi et al., 2024; Ou et al., 2024). Subsequent work has explored scaling properties (Nie et al., 2024), including training from scratch and adaptation from pre-trained autoregressive models (Nie et al., 2025; Gong et al., 2024; Ye et al., 2025).

**Alignment of MDMs.** Recent studies have emerged to explore aligning MDMs. Zekri and Boullé (2025) introduced a general policy-gradient method leveraging the denoising distribution of the discrete diffusion model during the reverse process. Borso et al. (2025) adopts a continuous-time Markov chain view for discrete diffusion, treating each token step as an action, and introduces a DPO variant, validated on small-scale binary sequence generation. Zhao et al. (2025); Yang et al. (2025); Tang et al. (2025) treat each token step as an action and develop GRPO-based methods to enhance reasoning ability. Huang et al. (2025) propose a GRPO variant viewing intermediate diffusion steps as the RL trajectory, focusing on the reasoning and code generation tasks. Gong et al. (2025) present a GRPO-based algorithm for code generation with a coupled-sampling variance-reduction technique, which can be used complementary to VRPO. Compared with these existing and concurrent works, we investigate the alignment of MDMs based on DPO with ELBO-based log-likelihood approximation, which serves as a natural choice for diffusion models. The proposed VRPO incorporates theoretically grounded variance-reduction techniques and is validated through large-scale experiments on general alignment tasks beyond reasoning and code generation. We believe our work provides a meaningful complement to existing MDMs alignment methods.

**Variance reduction techniques.** Our work relates to the broad fields of variance reduction in Monte Carlo methods, doubly stochastic optimization, and variational inference. In Monte Carlo methods, variance reduction aims to enhance estimation accuracy by improving sampling strategies. Classic techniques include control variables and stratified sampling (Kroese et al., 2013), where our approach adapts antithetic variates to couple correlated ELBO terms. The doubly expectation in ELBOs further parallels the nested structure in doubly SGD (Dai et al., 2014; Titsias and Lázaro-Gredilla, 2014; Gower et al., 2020; Kim et al., 2024), motivating decomposition via the law of total variance to isolate distinct variance sources. Our approach also conceptually aligns with importance weighted variational inference (Burda et al., 2016; Huang and Courville, 2019), where the outer bias is decreased by reducing the inner variance.

## 6 CONCLUSION

We analyze the challenges of aligning MDMs with human preference, particularly the high variance and bias inherent in the ELBO-based likelihood estimation. To address these issues, we propose VRPO, a systematic framework that incorporates variance reduction techniques with both theoretical guarantees and empirical validation, which provides transferable insight beyond specific architectures or datasets. The resulting model, LLaDA 1.5, demonstrates stronger general capabilities than LLaDA, with pronounced strengths in mathematics, coding, and alignment, supporting the effectiveness of VRPO at a large scale. Careful ablation studies further investigate each component in VRPO, showing their effect on variance reduction and thus the stability and efficiency of the optimization. Potential extensions of the proposed variance reduction techniques to broader RL-based alignment algorithms are also discussed. We hope this work provides useful guidance for future research on MDM alignment and contributes to the continued development of diffusion-based language models.

ETHICS STATEMENT

This paper focuses on aligning MDMs with human preferences to improve helpfulness. Nonetheless, misuse risks remain: the models may still generate discriminatory, biased, or otherwise harmful content. To mitigate these risks, we curated and filtered the preference data to remove harmful material where feasible and will continue to evaluate and refine our safeguards to reduce harmful outputs.

REPRODUCIBILITY STATEMENT

We provide complete proofs of our theoretical results in Section 3 and Appendix C. For experiments, Appendix E details the setup, including model architectures and configurations, data processing steps, training hyperparameters, evaluation metrics, and the configurations of our ablation studies. We will release LLaDA 1.5 after the blind review period.

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

# Contents

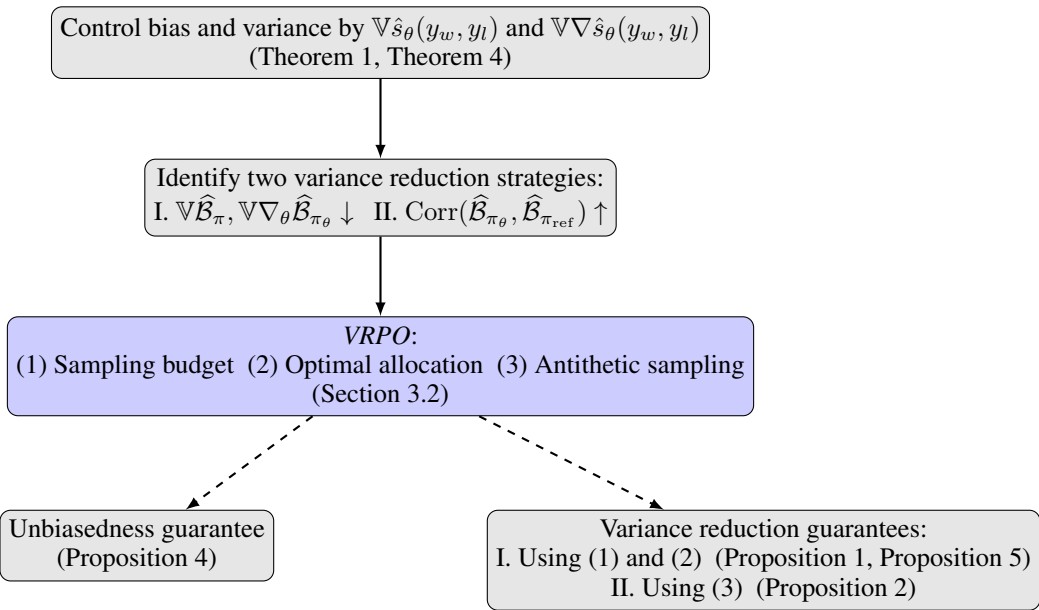

Figure 4: **Illustration of the analysis process.** This diagram outlines the conceptual flow that leads to the proposed VRPO method. Gray boxes represent theoretical analyses, and the blue box highlights the final sampling strategy. Starting from a bias and variance analysis of the estimated loss and gradient, we identify the score-estimator variance as a dominant controller. These theoretical findings collectively motivate the design of the VRPO algorithm, which is equipped with provable properties (dashed lines): unbiasedness and guaranteed variance reduction.

## A  THE USE OF LARGE LANGUAGE MODELS

During the preparation of this manuscript, we used large language models to assist with vocabulary selection, identify awkward or ungrammatical sentences, and revise the text for clarity and readability.

## B  ADDITIONAL FORMULATION OF MDMS

For completeness, we introduce the formulation of MDMs in this section. For further details, please refer to prior works (Shi et al., 2024; Sahoo et al., 2024; Ou et al., 2024).

In the forward process, MDMs corrupt an original sequence $y$ by progressively masking tokens. Each token is masked independently at a noise level $t \in [0, 1]$. Let $y \in \{0, 1, \ldots, K-1\}^L$ be the original full response, where $K$ denotes the vocabulary size and $L$ denotes the sequence length, given a prompt $x$, the forward process is formulated as:

$$q(y_t|t, y, x) = \prod_{i=1}^{L} q(y_t^i|t, y^i, x), \quad q(y_t^i|t, y^i, x) = \begin{cases} 1-t, & y_t^i = y^i, \\ t, & y_t^i = \mathbf{M}, \end{cases} \tag{10}$$

where $y^i$ denotes the $i$-th token of response $y$, and $\mathbf{M}$ denotes the mask token.

The reverse process starts from a fully masked sequence and gradually unmask tokens to recover meaningful language sequences. For timesteps $0 \leq s < t \leq 1$, the reverse process is defined as:

$$q(y_s|s, t, y_t, x) = \prod_{i=1}^{L} q(y_s^i|s, t, y_t, x), \quad q(y_s^i|s, t, y_t, x) = \begin{cases} \frac{t-s}{t} p_\theta(y^i|y_t, x), & y_t^i = \mathbf{M} \wedge y_s^i \neq \mathbf{M}, \\ \frac{s}{t}, & y_t^i = \mathbf{M} \wedge y_s^i = \mathbf{M}, \\ 1, & y_t^i \neq \mathbf{M} \wedge y_s^i = y_t^i, \\ 0, & \text{otherwise}, \end{cases} \tag{11}$$

where $p_\theta$ is modeled by the mask prediction model.

As stated in Section 2.2, the exact log-likelihood $\log \pi(y|x)$ in MDMs is typically approximated by its ELBO (Lou et al., 2023; Ou et al., 2024; Shi et al., 2024; Sahoo et al., 2024):

$$\mathcal{B}_\pi(y|x) \triangleq \mathbb{E}_{t \sim \mathcal{U}[0,1]} \mathbb{E}_{y_t \sim q(y_t|t,y,x)} \ell_\pi(y_t, t, y|x), \tag{12}$$

where

$$\ell_\pi(y_t, t, y|x) \triangleq \left[ \frac{1}{t} \sum_{i=1}^{L} \mathbf{1}[y_t^i = \mathbf{M}] \log p_\theta(y^i|y_t, x) \right]. \tag{13}$$

As noted in (Ou et al., 2024; Nie et al., 2025), the following formulation is an equivalent approximation:

$$\mathcal{B}'_\pi(y|x) \triangleq \mathbb{E}_{l \sim \mathcal{U}(\{1,2,...,L\})} \mathbb{E}_{y_l \sim q(y_l|l,y,x)} \ell'_\pi(y_l, l, y|x), \tag{14}$$

where

$$\ell'_\pi(y_l, l, y|x) \triangleq \left[ \frac{L}{l} \sum_{i=1}^{L} \mathbf{1}[y_l^i = \mathbf{M}] \log p_\theta(y^i|y_l, x) \right], \tag{15}$$

with $l$ uniformly sampled from $\{1, 2, \ldots, L\}$, and $y_l$ denoting the sequence obtained by masking $l$ tokens without replacement.

In practice, although Eq. (14) and Eq. (12) are equivalent in expectation (Ou et al., 2024), the former typically yields lower variance during estimation (Nie et al., 2025). Intuitively, Eq. (14) deterministically masks exactly $l$ out of $L$ tokens in each sequence, providing more consistent samples. In contrast, Eq. (12) relies on masking an expected fraction $t$ of the tokens, which introduces greater variability into the estimation process. In practice, we apply Eq. (14) as our log-likelihood estimator.

## C  ADDITIONAL THEORETICAL CONTENTS

**Notations.** We use $\mathcal{S}_{\mathcal{B}_{\pi|y}}$ and $\mathcal{S}_{\hat{s}|y_w, y_l}$ to denote the stochastic sampling in the ELBO estimates and the resulting preference score, respectively. Let $\mathcal{S}_t$ and $\mathcal{S}_{y_{t(j)}|y}$ be as defined in Eq. (5), $\mathcal{S}_{\text{data}}$ be as defined in Eq. (8). $\Theta(\cdot)$ denotes functions of the same order.

### C.1  AUXILIARY LEMMAS

#### C.1.1  PROPERTIES OF $\log \sigma(\cdot)$ (LEMMA 1)

**Lemma 1** (Properties of $\log \sigma(x)$). *Let $f(x) = \log \sigma(x)$, where $x \in \mathbb{R}$ and $\sigma(x) = \frac{1}{1+e^{-x}}$ denotes the sigmoid function. Then $f$ satisfies the following properties:*

  *(i) **concavity:** $f(x)$ is concave;*

 *(ii) **continuity:** $f$ is 1-Lipschitz continuous on $\mathbb{R}$, i.e., for all $x_1, x_2 \in \mathbb{R}$,*
$$|f(x_1) - f(x_2)| \leq |x_1 - x_2|;$$

*(iii) **smoothness:** $f$ is $\frac{1}{4}$-smooth on $\mathbb{R}$, i.e., its derivative is $\frac{1}{4}$-Lipschitz continuous: for all $x_1, x_2 \in \mathbb{R}$,*
$$|f'(x_1) - f'(x_2)| \leq \frac{1}{4}|x_1 - x_2|.$$

*Proof.* We first compute the first and second derivatives of $f$. Note that

$$f'(x) = \frac{e^{-x}}{1 + e^{-x}} = \frac{1}{1 + e^x} \in (0, 1),$$

and

$$f''(x) = -\frac{e^x}{(1 + e^x)^2} \in [-\frac{1}{4}, 0).$$

**(i)** Since $f''(x) \geq 0$ for all $x \in \mathbb{R}$, we have $f$ is concave.

**(ii)** We observe from above that $|f'(x)| \leq 1$ for all $x \in \mathbb{R}$, implying that $f$ is 1-Lipschitz continuous.

**(iii)** Since $|f''(x)| \leq \frac{1}{4}$ for all $x \in \mathbb{R}$, the derivative $f'(x)$ is $\frac{1}{4}$-Lipschitz continuous, and thus $f$ is $\frac{1}{4}$-smooth. $\qquad\square$

### C.1.2 INTERCHANGEABILITY OF EXPECTATION AND GRADIENT (LEMMA 2)

**Lemma 2** (Interchangeability of expectation and gradient). *Let $\theta \in \mathbb{R}^d$, and let $X$ be a random variable (or random vector) taking values in a measurable space $\mathcal{X}$. Suppose $f_\theta : \mathbb{R}^d \times \mathcal{X} \to \mathbb{R}$ is differentiable with respect to $\theta$ for all $X \in \mathcal{X}$, and there exists a constant $C > 0$ such that $\|\nabla_\theta f_\theta(X)\|_2 \leq C$ for all $X \in \mathcal{X}$. Then the expectation and gradient operators are interchangeable:*

$$\nabla_\theta \mathbb{E} f_\theta(X) = \mathbb{E} \nabla_\theta f_\theta(X).$$

*Proof.* Let $\theta \in \mathbb{R}^d$ be fixed. For all $X$, for each $i \in \{1, \ldots, d\}$, define $g_i(X) := \frac{\partial}{\partial \theta_i} f_\theta(X)$, which exists since $f_\theta(X)$ is differentiable w.r.t. $\theta$. By assumption, we have

$$|g_i(X)| \leq \|\nabla_\theta f_\theta(X)\|_2 \leq C.$$

For each $i$, by the mean value theorem and dominated convergence theorem (Bartle, 2014, Chapter 5), we can interchange the expectation and derivative:

$$\frac{\partial}{\partial \theta_i} \mathbb{E} f_\theta(X) = \mathbb{E} \frac{\partial}{\partial \theta_i} f_\theta(X).$$

Applying this for each coordinate and stacking the results gives the full gradient interchangeability:

$$\nabla_\theta \mathbb{E} f_\theta(X) = \mathbb{E} \nabla_\theta f_\theta(X).$$

$\qquad\square$

### C.1.3 BIAS AND VARIANCE OF TRANSFORMED RANDOM VARIABLE (LEMMA 3)

**Lemma 3** (Bias and variance of transformed random variable). *Let $X_\theta$ be a real-valued random variable with $\mathbb{E} X_\theta = \mu_\theta$ with parameter $\theta \in \mathbb{R}^d$, and define function $f(x) = \log \sigma(x)$ on $\mathbb{R}$, where $\sigma(x) = \frac{1}{1+e^{-x}}$ denotes the sigmoid function. Then:*

*(i) The transformed random variable satisfies:*

$$\mathbb{E}|f(X_\theta) - f(\mu_\theta)| \leq \sqrt{\mathbb{V} X_\theta}, \tag{16}$$

$$\mathbb{V} f(X_\theta) \leq 4 \mathbb{V} X_\theta. \tag{17}$$

*(ii) Suppose there exists a constant $C \geq 0$ such that the gradient of $X_\theta$ is uniformly bounded as $\|\nabla_\theta X_\theta\|_2 \leq C$. Then, the gradient satisfies:*

$$\mathbb{E}\|\nabla_\theta f(X_\theta) - \nabla_\theta f(\mu_\theta)\|_2 \leq \frac{C}{4} \sqrt{\mathbb{V} X_\theta} + \sqrt{\operatorname{tr}\mathbb{V}\nabla_\theta X_\theta}, \tag{18}$$

$$\operatorname{tr}\mathbb{V}\nabla_\theta f(X_\theta) \leq \frac{C^2}{8} \mathbb{V} X_\theta + \operatorname{tr}\mathbb{V}\nabla_\theta X_\theta. \tag{19}$$

*Proof.* *(i)* As $f = \log \sigma$ is 1-Lipschitz continuous by Lemma 1, for Eq. (16), we have:

$$
\begin{aligned}
\mathbb{E}|f(X_\theta) - f(\mu_\theta)| &\leq \mathbb{E}|X_\theta - \mu_\theta| \\
&= \mathbb{E}\sqrt{(X_\theta - \mu_\theta)^2} \\
&\leq \sqrt{\mathbb{E}(X_\theta - \mu_\theta)^2} && \text{(Jensen's inequality)} \\
&= \sqrt{\mathbb{V} X_\theta} && (\mathbb{E} X_\theta = \mu_\theta)
\end{aligned}
$$

For Eq. (17), we have:

$$\mathbb{V}f(X_\theta) = \mathbb{E}\big(f(X_\theta) - \mathbb{E}f(X_\theta)\big)^2$$

$$\leq \mathbb{E}\Big(\big|f(X_\theta) - f(\mathbb{E}X_\theta)\big| + \big|f(\mathbb{E}X_\theta) - \mathbb{E}f(X_\theta))\big|\Big)^2 \qquad \text{(triangle inequality)}$$

$$\leq 2\mathbb{E}\big(f(X_\theta) - f(\mathbb{E}X_\theta)\big)^2 + \mathbb{E}\big(f(\mathbb{E}X_\theta) - \mathbb{E}f(X_\theta))\big)^2 \qquad ((a+b)^2 \leq 2(a^2+b^2))$$

$$= 2\mathbb{E}\big(f(X_\theta) - f(\mathbb{E}X_\theta)\big)^2 + \big(f(\mathbb{E}X_\theta) - \mathbb{E}f(X_\theta))\big)^2$$

$$= 2\mathbb{E}\big(f(X_\theta) - f(\mathbb{E}X_\theta)\big)^2 + \Big(\mathbb{E}\big(f(\mathbb{E}X_\theta) - f(X_\theta))\big)\Big)^2$$

$$\leq 2\mathbb{E}\big(f(X_\theta) - f(\mathbb{E}X_\theta)\big)^2 + \mathbb{E}\big(f(\mathbb{E}X_\theta) - f(X_\theta)\big)^2 \qquad \text{(Jensen's inequality)}$$

$$= 4\mathbb{E}\big(f(X_\theta) - f(\mathbb{E}X_\theta)\big)^2$$

$$\leq 4\mathbb{E}(X_\theta - \mathbb{E}X_\theta)^2 \qquad (f \text{ is 1-Lipschitz continuous by Lemma 1})$$

$$= 4\mathbb{V}X_\theta$$

*(ii)* Using the chain rule and the bounded gradient assumption, for Eq. (18), we have

$$\mathbb{E}\big\|\nabla_\theta f(X_\theta) - \nabla_\theta f(\mu_\theta)\big\|_2$$

$$= \mathbb{E}\big\|f'(X_\theta)\nabla_\theta X_\theta - f'(\mu_\theta)\nabla_\theta \mu_\theta\big\|_2$$

$$\leq \mathbb{E}\big\|\big(f'(X_\theta) - f'(\mu_\theta)\big)\nabla_\theta X_\theta\big\|_2 + \mathbb{E}\big\|f'(\mu_\theta)\big(\nabla_\theta X_\theta - \nabla_\theta \mu_\theta\big)\big\|_2 \qquad \text{(triangle inequality)}$$

$$= \mathbb{E}[|f'(X_\theta) - f'(\mu_\theta)| \cdot \|\nabla_\theta X_\theta\|_2] + |f'(\mu_\theta)| \cdot \mathbb{E}\|\nabla_\theta X_\theta - \nabla_\theta \mu_\theta\|_2$$

$$\leq C \cdot \mathbb{E}\big|f'(X_\theta) - f'(\mu_\theta)\big| + \mathbb{E}\|\nabla_\theta X_\theta - \nabla_\theta \mu_\theta\|_2 \qquad (f \text{ is 1-Lipschitz continuous by Lemma 1})$$

$$\leq \frac{C}{4} \cdot \mathbb{E}|X_\theta - \mu_\theta| + \mathbb{E}\|\nabla_\theta X_\theta - \nabla_\theta \mu_\theta\|_2 \qquad (f \text{ is } \tfrac{1}{4}\text{-Lipschitz smooth by Lemma 1})$$

$$= \frac{C}{4} \cdot \mathbb{E}|X_\theta - \mu_\theta| + \mathbb{E}\|\nabla_\theta X_\theta - \mathbb{E}\nabla_\theta X_\theta\|_2 \qquad (\mathbb{E}X_\theta = \mu_\theta \text{ and Lemma 2})$$

$$\leq \frac{C}{4}\sqrt{\mathbb{E}(X_\theta - \mu_\theta)^2} + \sqrt{\mathbb{E}\|\nabla_\theta X_\theta - \mathbb{E}\nabla_\theta X_\theta\|_2^2} \qquad \text{(Jensen's inequality)}$$

$$= \frac{C}{4}\sqrt{\mathbb{V}X_\theta} + \sqrt{\text{tr}\mathbb{V}\nabla_\theta X_\theta}.$$

To prove Eq. (19), we begin by decomposing the variance of the estimated gradient into three terms:

$$\text{tr}\mathbb{V}\nabla_\theta f(X_\theta) = \mathbb{E}\big\|\nabla_\theta f(X_\theta) - \mathbb{E}\nabla_\theta f(X_\theta)\big\|_2^2 = \mathbb{E}\big\|f'(X_\theta)\nabla_\theta X_\theta - \mathbb{E}\left[f'(X_\theta)\nabla_\theta X_\theta\right]\big\|_2^2$$

$$\leq \underbrace{\mathbb{E}\big\|f'(X_\theta)\nabla_\theta X_\theta - f'(\mathbb{E}X_\theta)\nabla_\theta X_\theta\big\|_2^2}_{\text{(I)}} + \underbrace{\mathbb{E}\big\|f'(\mathbb{E}X_\theta)\nabla_\theta X_\theta - f'(\mathbb{E}X_\theta)\mathbb{E}\nabla_\theta X_\theta\big\|_2^2}_{\text{(II)}}$$

$$+ \underbrace{\mathbb{E}\big\|f'(\mathbb{E}X_\theta)\mathbb{E}\nabla_\theta X_\theta - \mathbb{E}\left[f'(X_\theta)\nabla_\theta X_\theta\right]\big\|_2^2}_{\text{(III)}}.$$

We now bound each term separately.

Term (I). Using the bounded gradient assumption $\|\nabla_\theta X_\theta\|_2 \leq C$ and the $\frac{1}{4}$-Lipschitz smoothness of $f$ (by Lemma 1), we have:

$$\text{(I)} = \mathbb{E}\Big[\big|f'(X_\theta) - f'(\mathbb{E}X_\theta)\big|^2 \cdot \|\nabla_\theta X_\theta\|_2^2\Big]$$

$$\leq C^2\mathbb{E}\big|f'(X_\theta) - f'(\mathbb{E}X_\theta)\big|^2$$

$$\leq \frac{C^2}{16}\mathbb{E}|X_\theta - \mathbb{E}X_\theta|^2 = \frac{C^2}{16}\mathbb{V}X_\theta.$$

Term (II). Since $f'$ is bounded by 1 (by Lemma 1), we have:

$$
\begin{aligned}
\text{(II)} &= \left|f'(\mathbb{E}X_\theta)\right|^2 \cdot \mathbb{E}\|\nabla_\theta X_\theta - \mathbb{E}\nabla_\theta X_\theta\|_2^2 \\
&\leq \text{tr}\mathbb{V}\nabla_\theta X_\theta.
\end{aligned}
$$

Term (III). Applying Jensen's inequality and again using the smoothness of $f$ and boundedness of $\nabla_\theta X_\theta$, we have:

$$
\begin{aligned}
\text{(III)} &= \left\|f'(\mathbb{E}X_\theta)\mathbb{E}\nabla_\theta X_\theta - \mathbb{E}f'(X_\theta)\nabla_\theta X_\theta\right\|_2^2 \\
&= \left\|\mathbb{E}\left[f'(\mathbb{E}X_\theta)\nabla_\theta X_\theta - f'(X_\theta)\nabla_\theta X_\theta\right]\right\|_2^2 \\
&\leq \mathbb{E}\left\|f'(\mathbb{E}X_\theta)\nabla_\theta X_\theta - f'(X_\theta)\nabla_\theta X_\theta\right\|_2^2 \\
&= \mathbb{E}\left[\left|f'(\mathbb{E}X_\theta) - f'(X_\theta)\right|^2 \cdot \|\nabla_\theta X_\theta\|_2^2\right] \\
&\leq C^2\mathbb{E}\left|f'(\mathbb{E}X_\theta) - f'(X_\theta)\right|^2 \leq \frac{C^2}{16}\mathbb{V}X_\theta.
\end{aligned}
$$

Summing all three terms yields:

$$
\text{tr}\mathbb{V}\nabla_\theta f(X_\theta) \leq \frac{C^2}{8}\mathbb{V}X_\theta + \text{tr}\mathbb{V}\nabla_\theta X_\theta.
$$

$\square$

### C.1.4 PREPARATION FOR TIGHTNESS ANALYSIS (LEMMA 4)

**Lemma 4.** *Assume that a random variable $X$ has finite mean, variance, and kurtosis, i.e., $\mathbb{E}[X] < \infty$, $\mathbb{V}[X] < \infty$, and $\kappa \triangleq \frac{\mathbb{E}[(X-\mathbb{E}[X])^4]}{\mathbb{V}[X]^2} < \infty$. Then there exists a constant $c = \sqrt{0.2}(0.8)^2 \approx 0.2862$ such that:*

$$
\mathbb{E}[|X - \mathbb{E}[X]|] \geq \frac{c}{\kappa}\sqrt{\mathbb{V}[X]}.
$$

*Proof.* Let $\mu := \mathbb{E}[X]$, $\sigma := \sqrt{\mathbb{V}[X]}$, and define $Y := (X - \mu)^2$. Then,

$$
\mathbb{E}[Y] = \mathbb{V}[X] = \sigma^2 < \infty, \qquad \mathbb{E}[Y^2] = \mathbb{E}[(X - \mu)^4] = \kappa\sigma^4 < \infty.
$$

Applying the Paley–Zygmund inequality to the nonnegative random variable $Y$, we have: for any $0 \leq \theta \leq 1$:

$$
\mathbb{P}(Y \geq \theta\,\mathbb{E}[Y]) \geq \frac{(1-\theta)^2\,(\mathbb{E}[Y])^2}{\mathbb{E}[Y^2]} = \frac{(1-\theta)^2\,\sigma^4}{\kappa\sigma^4} = \frac{(1-\theta)^2}{\kappa}.
$$

Next, let $F_Y$ denote the cumulated density function of $Y$. Unrolling the expectation, we have:

$$
\mathbb{E}[|X - \mu|] = \mathbb{E}[\sqrt{Y}] = \int_0^\infty \sqrt{y}\,dF_Y(y) \geq \int_{\theta\sigma^2}^\infty \sqrt{y}\,dF_Y(y) \geq \int_{\theta\sigma^2}^\infty \sqrt{\theta\sigma^2}\,dF_Y(y),
$$

$$
= \sqrt{\theta\sigma^2}\left[1 - F_Y(\theta\sigma^2)\right] = \sigma\sqrt{\theta}\mathbb{P}(Y \geq \theta\sigma^2) \geq \sigma\sqrt{\theta}\frac{(1-\theta)^2}{\kappa}.
$$

Maximizing the right hand side over $\theta \in [0, 1]$, we obtain

$$
\max_{0 \leq \theta \leq 1} \sqrt{\theta}(1-\theta)^2 = \sqrt{0.2}(1 - 0.2)^2.
$$

Letting $c = \sqrt{0.2}(0.8)^2$, we conclude that

$$\mathbb{E}[|X - \mu|] \geq \frac{c}{\kappa}\sigma.$$

$\square$

### C.1.5 VARIANCE OF ELBO ESTIMATOR (LEMMA 5)

**Lemma 5** (Variance of ELBO estimator). *Letting $\widehat{\mathcal{B}}_\pi(y)$ be as defined in Eq. (6), we have:*

*(i) The variance of the ELBO estimator satisfies:*

$$\mathbb{V}\widehat{\mathcal{B}}_\pi(y) = \frac{1}{n_t}\underbrace{\mathbb{V}_t\mathbb{E}_{y_t|t,y}\ell_\pi(y_t,t,y)}_{\triangleq V_t} + \frac{1}{n_t n_{y_t}}\underbrace{\mathbb{E}_t\mathbb{V}_{y_t|t,y}\ell_\pi(y_t,t,y)}_{\triangleq V_{y_t}}.$$

*(ii) The variance of the gradient of the ELBO estimator for the model policy $\pi_\theta$ satisfies:*

$$\mathbb{V}\nabla_\theta\widehat{\mathcal{B}}_{\pi_\theta}(y) = \frac{1}{n_t}\underbrace{\mathbb{V}_t\mathbb{E}_{y_t|t,y}\nabla_\theta\ell_{\pi_\theta}(y_t,t,y)}_{\triangleq V_t^\nabla} + \frac{1}{n_t n_{y_t}}\underbrace{\mathbb{E}_t\mathbb{V}_{y_t|t,y}\nabla_\theta\ell_{\pi_\theta}(y_t,t,y)}_{\triangleq V_{y_t}^\nabla}.$$

*The $V_t$ (or $V_t^\nabla$) and $V_{y_t}$ (or $V_{y_t}^\nabla$) capture variance across timesteps and variance due to the noise at each step, which are inherently determined by the data and the forward process and cannot be reduced.*

*Proof.* For **(i)**, by the law of total variance,

$$\mathbb{V}\widehat{\mathcal{B}}_\pi(y) = \underbrace{\mathbb{V}_{\mathcal{S}_t}\mathbb{E}_{\{\mathcal{S}_{y_{t(j)}|y}\}_{j=1}^{n_{y_t}}|\mathcal{S}_t}\widehat{\mathcal{B}}_\pi(y)}_{(\mathrm{I})} + \underbrace{\mathbb{E}_{\mathcal{S}_t}\mathbb{V}_{\{\mathcal{S}_{y_{t(j)}|y}\}_{j=1}^{n_{y_t}}|\mathcal{S}_t}\widehat{\mathcal{B}}_\pi(y)}_{(\mathrm{II})}.$$

Term (I). Conditioned on the $t$-sample, the inner expectation is:

$$\mathbb{E}_{\{\mathcal{S}_{y_{t(j)}|y}\}_{j=1}^{n_{y_t}}|\mathcal{S}_t}\widehat{\mathcal{B}}_\pi(y) = -\frac{1}{n_t}\sum_{j=1}^{n_t}\mathbb{E}_{\mathcal{S}_{y_{t(j)}|y}|\mathcal{S}_t}\frac{1}{n_{y_t}}\sum_{k=1}^{n_{y_t}}\ell_\pi(y_{t(j)}^{(k)},t^{(j)},y) = -\frac{1}{n_t}\sum_{j=1}^{n_t}\mathbb{E}_{y_t|t^{(j)},y}\ell_\pi(y_t,t^{(j)},y).$$

Since terms in $\mathcal{S}_t$ are i.i.d. sampled, the outer variance is:

$$(\mathrm{I}) = \mathbb{V}_{\mathcal{S}_t}\mathbb{E}_{\{\mathcal{S}_{y_{t(j)}|y}\}_{j=1}^{n_{y_t}}|\mathcal{S}_t}\widehat{\mathcal{B}}_\pi(y) = -\frac{1}{n_t^2}\mathbb{V}_{\mathcal{S}_t}\sum_{j=1}^{n_t}\mathbb{E}_{y_t|t^{(j)},y}\ell_\pi(y_t,t,y) = -\frac{1}{n_t}\mathbb{V}_t\mathbb{E}_{y_t|t,y}\ell_\pi(y_t,t,y).$$

Term (II). Conditioned on the $t$-sample, the inner variance is:

$$\mathbb{V}_{\{\mathcal{S}_{y_{t(j)}|y}\}_{j=1}^{n_{y_t}}|\mathcal{S}_t}\widehat{\mathcal{B}}_\pi(y) = -\frac{1}{n_t^2}\sum_{j=1}^{n_t}\mathbb{V}_{\mathcal{S}_{y_{t(j)}|y}|\mathcal{S}_t}\frac{1}{n_{y_t}}\sum_{k=1}^{n_{y_t}}\ell_\pi(y_{t(j)}^{(k)},t^{(j)},y)$$

$$= -\frac{1}{n_t^2}\sum_{j=1}^{n_t}\frac{1}{n_{y_t}^2}\sum_{k=1}^{n_{y_t}}\mathbb{V}_{y_t|t^{(j)},y}\ell_\pi(y_t,t^{(j)},y) = -\frac{1}{n_t^2 n_{y_t}}\sum_{j=1}^{n_t}\mathbb{V}_{y_t|t^{(j)},y}\ell_\pi(y_t,t^{(j)},y).$$

Taking the expectation over $\mathcal{S}_t$ yields:

$$\mathbb{E}_{\mathcal{S}_t}\mathbb{V}_{\mathcal{S}_{y_t}|\mathcal{S}_t}\widehat{\mathcal{B}}_\pi(y) = -\frac{1}{n_t^2 n_{y_t}}\mathbb{E}_{\mathcal{S}_t}\sum_{j=1}^{n_t}\mathbb{V}_{y_t|t^{(j)},y}\ell_\pi(y_t,t^{(j)},y) = -\frac{1}{n_t n_{y_t}}\mathbb{E}_t\mathbb{V}_{y_t|t,y}\ell_\pi(y_t,t,y).$$

Combining (I) and (II) gives the result:

$$\mathbb{V}\widehat{\mathcal{B}}_\pi(y) = \frac{1}{n_t}V_t + \frac{1}{n_t n_{y_t}}V_{y_t}.$$

For **(ii)**, as $\nabla_\theta\widehat{\mathcal{B}}_{\pi_\theta}(y)$ has similar structure as $\widehat{\mathcal{B}}_{\pi_\theta}(y)$:

$$\nabla_\theta\widehat{\mathcal{B}}_{\pi_\theta}(y) \triangleq \frac{1}{n_t}\sum_{j=1}^{n_t}\frac{1}{n_{y_t}}\sum_{k=1}^{n_{y_t}}\nabla_\theta\ell_{\pi_\theta}(y_{t(j)}^{(k)},t^{(j)},y),$$

the proof closely follows that for **(i)**, and thus we omit the details here. $\square$

## C.2 BIAS AND VARIANCE OF ESTIMATED LOSS

### C.2.1 UNBIASEDNESS OF PREFERENCE SCORE ESTIMATOR (PROPOSITION 3)

**Proposition 3** (Unbiasedness of preference score estimator). *The preference score estimator defined in Eq.* (8) *is an unbiased estimator of the true preference score defined in Eq.* (7):

$$\mathbb{E}_{\mathcal{S}_{\hat{s}|y_w,y_l}}\left[\hat{s}_\theta(y_w,y_l)\right] = s_\theta(y_w,y_l).$$

*Proof.* First, by the i.i.d. sampling of timesteps and masked data, i.e.,

$$\mathcal{S}_t \triangleq \{t^{(j)}\}_{j=1}^{n_t} \overset{\text{i.i.d.}}{\sim} \mathcal{U}[0,1] \quad \text{and} \quad \mathcal{S}_{y_{t(j)}|y} \triangleq \{y_{t^{(j)}}^{(k)}\}_{k=1}^{n_{y_t}} \overset{\text{i.i.d.}}{\sim} q(y_t \mid t^{(j)}, y), \; j = 1, \ldots, n_t,$$

and $\mathcal{S}_{y_{t(j)}|y} \perp\!\!\!\perp \mathcal{S}_{y_{t(j')}|y}$ for $j \neq j'$, the ELBO estimator (Eq. (6)) is unbiased:

$$\mathbb{E}_{\mathcal{S}_t,\{\mathcal{S}_{y_{t(j)}|y}\}_{j=1}^{n_t}}\widehat{\mathcal{B}}_\pi(y) = \mathbb{E}_{\mathcal{S}_t}\frac{1}{n_t}\sum_{j=1}^{n_t}\mathbb{E}_{\mathcal{S}_{y_{t(j)}|y}}\frac{1}{n_{y_t}}\sum_{k=1}^{n_{y_t}}\ell_\pi(y_{t^{(j)}}^{(k)}, t^{(j)}, y)$$

$$= \mathbb{E}_{t\sim\mathcal{U}[0,1]}\mathbb{E}_{y_t\sim q(y_t|t,y)}\ell_\pi(y_t,t,y \mid x) = \mathcal{B}_\pi(y).$$

Since the preference score estimator is a linear combination of four ELBO estimators, by the linearity of the expectation, we have:

$$\mathbb{E}\left[\hat{s}_\theta(y_w,y_l)\right] = \beta\mathbb{E}\left[\widehat{\mathcal{B}}_{\pi_\theta}(y_w)\right] - \beta\mathbb{E}\left[\widehat{\mathcal{B}}_{\pi_{\text{ref}}}(y_w)\right] - \beta\mathbb{E}\left[\widehat{\mathcal{B}}_{\pi_\theta}(y_l)\right] + \beta\mathbb{E}\left[\widehat{\mathcal{B}}_{\pi_{\text{ref}}}(y_l)\right]$$

$$= \beta\left(\mathcal{B}_{\pi_\theta}(y_w) - \mathcal{B}_{\pi_{\text{ref}}}(y_w)\right) - \beta\left(\mathcal{B}_{\pi_\theta}(y_l) - \mathcal{B}_{\pi_{\text{ref}}}(y_l)\right) = s_\theta(y_w,y_l).$$

□

### C.2.2 EFFECT OF PREFERENCE SCORE ESTIMATOR VARIANCE (THEOREM 1)

**Theorem 1.** *Given a pair of preference data $y_w, y_l$, the bias and variance of $\widehat{\ell}_{\text{DPO}-\text{E}}(y_w, y_l; \theta)$ over stochastic sampling in the score estimation can be bounded as:*

$$\mathbb{E}_{\mathcal{S}_{\hat{s}|y_w,y_l}}\left[\left|\ell_{\text{DPO}-\text{E}}(y_w,y_l;\theta) - \widehat{\ell}_{\text{DPO}-\text{E}}(y_w,y_l;\theta)\right|\right] \leq \sqrt{\mathbb{V}_{\mathcal{S}_{\hat{s}|y_w,y_l}}\left[\hat{s}_\theta(y_w,y_l)\right]},$$

$$\mathbb{V}_{\mathcal{S}_{\hat{s}|y_w,y_l}}\left[\widehat{\ell}_{\text{DPO}-\text{E}}(y_w,y_l;\theta)\right] \leq 4\mathbb{E}_{y_w,y_l}\left[\mathbb{V}_{\mathcal{S}_{\hat{s}|y_w,y_l}}\left[\hat{s}_\theta(y_w,y_l)\right]\right].$$

*Proof.* The proof is essentially based on the analysis of the bias and variance of the transformed random variable in Lemma 3.

By definitions in Eq. (7) and Eq. (8), we know that:

$$\mathbb{E}_{\mathcal{S}_{\hat{s}|y_w,y_l}}\left[\left|\ell_{\text{DPO}-\text{E}}(y_w,y_l;\theta) - \widehat{\ell}_{\text{DPO}-\text{E}}(y_w,y_l;\theta)\right|\right]$$

$$= \mathbb{E}_{\mathcal{S}_{\hat{s}|y_w,y_l}}\left[\left|\log\sigma\left(s_\theta(y_w,y_l)\right) - \log\sigma\left(\hat{s}_\theta(y_w,y_l)\right)\right|\right],$$

and

$$\mathbb{V}_{\mathcal{S}_{\hat{s}|y_w,y_l}}\left[\widehat{\ell}_{\text{DPO}-\text{E}}(y_w,y_l;\theta)\right] = \mathbb{V}_{\mathcal{S}_{\hat{s}|y_w,y_l}}\left[\log\sigma\left(\hat{s}_\theta(y_w,y_l)\right)\right].$$

According to Proposition 3, we know that $\hat{s}_\theta(y_w,y_l)$ is an unbiased estimator for $s_\theta(y_w,y_l)$ such that $\mathbb{E}_{\mathcal{S}_{\hat{s}|y_w,y_l}}\left[\hat{s}_\theta(y_w,y_l)\right] = s_\theta(y_w,y_l)$. Therefore, we can apply Lemma 3 presented previously to directly get the result.

□

### C.2.3 TIGHTNESS ANALYSIS (THEOREM 2, THEOREM 3)

**Theorem 2** (Tightness analysis of bias). *Assume that for any $y_w, y_l$, the estimator $\hat{s}_\theta(y_w, y_l)$ has finite mean, variance, and kurtosis, i.e., $\mathbb{E}_{S_{\hat{s}|y_w, y_l}}[\hat{s}_\theta(y_w, y_l)] < \infty$, $\mathbb{V}_{S_{\hat{s}|y_w, y_l}}[\hat{s}_\theta(y_w, y_l)] < \infty$, and $\kappa \triangleq \frac{\mathbb{E}[(\hat{s}_\theta(y_w, y_l) - \mathbb{E}[\hat{s}_\theta(y_w, y_l)])^4]}{\mathbb{V}[\hat{s}_\theta(y_w, y_l)]^2} < \infty$. Then, under a first-order Taylor expansion, the bias of $\widehat{\ell}_{\mathrm{DPO-E}}(y_w, y_l; \theta)$ scales proportionally to the square root of the variance of the score estimator as:*

$$\mathbb{E}_{\mathcal{S}_{\hat{s}|y_w, y_l}} \left[ \left| \ell_{\mathrm{DPO-E}}(y_w, y_l; \theta) - \widehat{\ell}_{\mathrm{DPO-E}}(y_w, y_l; \theta) \right| \right]$$

$$\approx \Theta \left( \mathbb{E}_{y_w, y_l} \left[ \left| f'\left( s_\theta(y_w, y_l) \right) \right| \sqrt{\mathbb{V}_{S_{\hat{s}|y_w, y_l}} \left[ \hat{s}_\theta(y_w, y_l) \right]} \right] \right),$$

*where $f(x) = \log \sigma(x)$, $f'(x) \in (0, 1)$.*

*Proof.* We omit the explicit conditioning on $y_w, y_l$ for brevity and denote $s_\theta := s_\theta(y_w, y_l)$, $\hat{s}_\theta := \hat{s}_\theta(y_w, y_l)$.

By a first-order Taylor expansion of $f(\hat{s}_\theta)$ around $s_\theta$, we have:

$$f(\hat{s}_\theta) = f(s_\theta) + f'(s_\theta)(\hat{s}_\theta - s_\theta) + O\left( (\hat{s}_\theta - s_\theta)^2 \right).$$

Ignoring the higher-order term yields the linear approximation:

$$f(\hat{s}_\theta) \approx f(s_\theta) + f'(s_\theta)(\hat{s}_\theta - s_\theta).$$

According to the definition, the bias of $\widehat{\ell}_{\mathrm{DPO-E}}(y_w, y_l; \theta)$ is:

$$\mathbb{E}_{\mathcal{S}_{\hat{s}|y_w, y_l}} \left[ \left| \ell_{\mathrm{DPO-E}}(y_w, y_l; \theta) - \widehat{\ell}_{\mathrm{DPO-E}}(y_w, y_l; \theta) \right| \right] = \mathbb{E}_{S_{\hat{s}|y_w, y_l}} \left[ \left| f\left( \hat{s}_\theta \right) - f\left( s_\theta \right) \right| \right].$$

Applying the linear approximation and using the fact that $f(s_\theta)$ is constant w.r.t. $S_{\hat{s}|y_w, y_l}$, we get:

$$\mathbb{E}_{S_{\hat{s}|y_w, y_l}} \left[ \left| f\left( \hat{s}_\theta \right) - f\left( s_\theta \right) \right| \right] \approx \mathbb{E}_{S_{\hat{s}|y_w, y_l}} \left[ \left| f'(s_\theta) \right| \left| \hat{s}_\theta - s_\theta \right| \right] = \left| f'(s_\theta) \right| \mathbb{E}_{S_{\hat{s}|y_w, y_l}} \left[ \left| \hat{s}_\theta - s_\theta \right| \right].$$

According to Jensen's inequality and by Proposition 3, which states that $\mathbb{E}_{S_{\hat{s}|y_w, y_l}}[\hat{s}_\theta] = s_\theta$, we have

$$\mathbb{E}_{S_{\hat{s}|y_w, y_l}} \left[ \left| \hat{s}_\theta - s_\theta \right| \right] \leq \sqrt{\mathbb{E}_{S_{\hat{s}|y_w, y_l}} \left[ \left( \hat{s}_\theta - s_\theta \right)^2 \right]} = \sqrt{\mathbb{V}_{S_{\hat{s}|y_w, y_l}} [\hat{s}_\theta]},$$

and according to Lemma 4, there exists a constant $c = \sqrt{0.2}(0.8)^2$ such that:

$$\mathbb{E}_{S_{\hat{s}|y_w, y_l}} \left[ \left| \hat{s}_\theta - s_\theta \right| \right] \geq \frac{c}{\kappa} \sqrt{\mathbb{V}_{S_{\hat{s}|y_w, y_l}} [\hat{s}_\theta]}.$$

Thus we get:

$$\frac{c}{\kappa} \sqrt{\mathbb{V}_{S_{\hat{s}|y_w, y_l}} [\hat{s}_\theta]} \leq \mathbb{E}_{S_{\hat{s}|y_w, y_l}} \left[ \left| \hat{s}_\theta - s_\theta \right| \right] \leq \sqrt{\mathbb{V}_{S_{\hat{s}|y_w, y_l}} [\hat{s}_\theta]},$$

which means:

$$\mathbb{E}_{\mathcal{S}_{\hat{s}|y_w, y_l}} \left[ \left| \ell_{\mathrm{DPO-E}}(y_w, y_l; \theta) - \widehat{\ell}_{\mathrm{DPO-E}}(y_w, y_l; \theta) \right| \right] \approx \left| f'(s_\theta) \right| \mathbb{E}_{S_{\hat{s}|y_w, y_l}} \left[ \left| \hat{s}_\theta - s_\theta \right| \right]$$

$$= \Theta \left( \left| f'(s_\theta) \right| \sqrt{\mathbb{V}_{S_{\hat{s}|y_w, y_l}} [\hat{s}_\theta]} \right).$$

Finally, from Lemma 1, we know $f'(s_\theta) \in (0, 1)$.

$\square$

*Remark* 1. The assumptions on $\hat{s}_\theta$ in Theorem 2, namely finite mean, variance, and kurtosis, are very mild and standard (Boucheron et al., 2003; Vershynin, 2018; Wainwright, 2019). These conditions exclude only extremely heavy-tailed distributions. They hold for all sub-Gaussian and sub-exponential distributions, specifically including Gaussian, uniform, exponential, and any bounded distributions. Since $\hat{s}_\theta$ is the estimated preference score computed from ELBOs derived using a neural network, it is naturally bounded in practice and thus satisfies these assumptions.

**Theorem 3** (Tightness analysis of variance). *Under a first-order Taylor expansion, the variance of* $\widehat{\ell}_{\mathrm{DPO-E}}(y_w, y_l; \theta)$ *scales proportionally to the variance of the score estimator as follows:*

$$\mathbb{V}_{S_{\hat{s}|y_w, y_l}}\left[\widehat{\ell}_{\mathrm{DPO-E}}(y_w, y_l; \theta)\right] \approx \left(f'\left(s_\theta(y_w, y_l)\right)\right)^2 \mathbb{V}_{S_{\hat{s}|y_w, y_l}}\left[\hat{s}_\theta(y_w, y_l)\right],$$

*where* $f(x) = \log \sigma(x)$, $f'(x) \in (0, 1)$.

*Proof.* We omit the explicit conditioning on $y_w, y_l$ for brevity and denote $s_\theta := s_\theta(y_w, y_l)$, $\hat{s}_\theta := \hat{s}_\theta(y_w, y_l)$.

By a first-order Taylor expansion of $f(\hat{s}_\theta)$ around $s_\theta$, we have:

$$f(\hat{s}_\theta) = f(s_\theta) + f'(s_\theta)(\hat{s}_\theta - s_\theta) + O\left((\hat{s}_\theta - s_\theta)^2\right).$$

Ignoring the higher-order term yields the linear approximation:

$$f(\hat{s}_\theta) \approx f(s_\theta) + f'(s_\theta)(\hat{s}_\theta - s_\theta).$$

According to the definition, the variance of $\widehat{\ell}_{\mathrm{DPO-E}}(y_w, y_l; \theta)$ is:

$$\mathbb{V}_{S_{\hat{s}|y_w, y_l}}\left[\widehat{\ell}_{\mathrm{DPO-E}}(y_w, y_l; \theta)\right] = \mathbb{V}_{S_{\hat{s}|y_w, y_l}}\left[f(\hat{s}_\theta)\right].$$

Applying the linear approximation and using the fact that $f(s_\theta)$ is constant w.r.t. $S_{\hat{s}|y_w, y_l}$, we get:

$$\mathbb{V}_{S_{\hat{s}|y_w, y_l}}\left[f(\hat{s}_\theta)\right] \approx \mathbb{V}_{S_{\hat{s}|y_w, y_l}}\left[f(s_\theta) + f'(s_\theta)(\hat{s}_\theta - s_\theta)\right] = \left(f'(s_\theta)\right)^2 \mathbb{V}_{S_{\hat{s}|y_w, y_l}}\left[\hat{s}_\theta\right].$$

Finally, from Lemma 1, we know $f'(s_\theta) \in (0, 1)$, ensuring the scaling factor is bounded. $\square$

### C.3 VARIANCE REDUCTION OF PREFERENCE SCORE ESTIMATOR

#### C.3.1 UNBIASEDNESS OF VRPO (PROPOSITION 4)

**Proposition 4** (Unbiasedness of VRPO). *Under the variance reduction techniques in VRPO (Section 3.2), the preference score estimator defined in Eq.* (8) *remains an unbiased estimator of the true preference score defined in Eq.* (7).

*Proof.* For *sampling budget* and *optimal allocation*, the proof of Proposition 3 for the unbiasedness of $\hat{s}_\theta(y_w, y_l)$ remains valid under variations in $n_t$ and $n_{y_t}$, so these do not affect the unbiasedness of the score estimator. For *antithetic sampling*, by linearity of expectation, the coupling of $\widehat{\mathcal{B}}_{\pi_\theta}(y)$ and $\widehat{\mathcal{B}}_{\pi_{\mathrm{ref}}}(y)$ also does not affect the unbiasedness of the score estimator.

$\square$

#### C.3.2 SAMPLING BUDGET AND ALLOCATION (PROPOSITION 1)

**Proposition 1** (Reduce the ELBO variance). *Given a total budget of* $n = n_t \times n_{y_t}$ *masked samples for estimating* $\widehat{\mathcal{B}}_\pi(y)$, *we have: (i)* $\mathbb{V}\widehat{\mathcal{B}}_\pi(y) = \Theta(\frac{1}{n})$, *(ii)* $\mathbb{V}\widehat{\mathcal{B}}_\pi(y)$ *is minimized when* $n_t = n, n_{y_t} = 1$.

*Proof.* The proof is essentially based on the variance analysis of the ELBO estimator in Lemma 5.

According to Lemma 5, we know that:

$$\mathbb{V}\widehat{\mathcal{B}}_\pi(y) = \frac{1}{n_t} V_t + \frac{1}{n_t n_{y_t}} V_{y_t}.$$

Given that $n = n_t \times n_{y_t}$ and a fixed allocation proportion $c \triangleq \frac{n_t}{n} \in [\frac{1}{n}, 1]$, we have:

$$\mathbb{V}\widehat{\mathcal{B}}_\pi(y) = \frac{1}{cn} V_t + \frac{1}{n} V_{y_t}.$$

Then, we have

$$\mathbb{V}\widehat{\mathcal{B}}_\pi(y) = \Theta(\frac{1}{n}),$$

and

$$\arg\min_{c \in [\frac{1}{n}, 1]} \mathbb{V}\widehat{\mathcal{B}}_\pi(y) = 1,$$

which gives the desired result.

$\square$

### C.3.3 ANTITHETIC SAMPLING (PROPOSITION 2)

**Proposition 2** (Increase the correlation). *Given any response $y$, supposing $\mathrm{Corr}\left(\widehat{\mathcal{B}}_{\pi_\theta}(y), \widehat{\mathcal{B}}_{\pi_{\mathrm{ref}}}(y)\right) >$ 0 when the Monte Carlo samples $S_t$ and $\{S_{y_{t(j)}|y}\}_{j=1}^{n_t}$ are shared between $\widehat{\mathcal{B}}_{\pi_\theta}(y)$ and $\widehat{\mathcal{B}}_{\pi_{\mathrm{ref}}}(y)$, we have: Sharing Monte Carlo samples yields lower $\mathbb{V}\hat{s}_\theta(y_w, y_l)$ than using independent samples.*

*Proof.* This result yields naturally from Eq. (9) that when $\mathrm{Corr}(\widehat{\mathcal{B}}_{\pi_\theta}(y), \widehat{\mathcal{B}}_{\pi_{\mathrm{ref}}}(y)) > 0$,

$$\mathbb{V}\widehat{\mathcal{B}}_{\pi_\theta}(y) + \mathbb{V}\widehat{\mathcal{B}}_{\pi_{\mathrm{ref}}}(y) - 2\mathrm{Corr}\left(\widehat{\mathcal{B}}_{\pi_\theta}(y), \widehat{\mathcal{B}}_{\pi_{\mathrm{ref}}}(y)\right)\sqrt{\mathbb{V}\widehat{\mathcal{B}}_{\pi_\theta}(y)\mathbb{V}\widehat{\mathcal{B}}_{\pi_{\mathrm{ref}}}(y)} < \mathbb{V}\widehat{\mathcal{B}}_{\pi_\theta}(y) + \mathbb{V}\widehat{\mathcal{B}}_{\pi_{\mathrm{ref}}}(y).$$

$\square$

### C.4 DEFERRED ANALYSIS OF ESTIMATED GRADIENT

In this section, we present a theoretical analysis of the effect of VRPO on gradient estimation, following a structure analogous to the loss analysis in the main paper.

We first introduce a bounded assumption on the gradient of per-step mask prediction loss $\ell_{\pi_\theta}$, which serves as a mild condition for the subsequent derivations.

**Assumption 1** (Bounded gradient of per-step mask prediction loss). The gradient of the per-step masked prediction loss $\ell_{\pi_\theta}(y_t, t, y)$ (Eq. (4)) is bounded, i.e., there exists a constant $0 \leq C < \infty$ such that $\|\nabla_\theta \ell_{\pi_\theta}(y_t, t, y)\|_2 \leq C$ for all $\theta$ in the model parameter space, $y$ in $\mathcal{D}$, and $t \in [0, 1]$.

This boundedness assumption is reasonable in practice and leads directly to the following corollary.

**Corollary 1** (Bounded gradient of preference score estimator). *Under Assumption 1, the gradient of the preference score estimator $\hat{s}_\theta(y_w, y_l)$ is bounded, i.e., there exists a constant $0 \leq \tilde{C} < \infty$ such that $\|\nabla_\theta \hat{s}_\theta(y_w, y_l)\|_2 \leq \tilde{C}$ for all $\theta$ in the model parameter space and $(y_w, y_l)$ in $\mathcal{D}$.*

*Proof.* Recall that the preference score estimator is defined as:

$$\hat{s}_\theta(y_w, y_l) = \beta\left(\widehat{\mathcal{B}}_{\pi_\theta}(y_w) - \widehat{\mathcal{B}}_{\pi_{\mathrm{ref}}}(y_w)\right) - \beta\left(\widehat{\mathcal{B}}_{\pi_\theta}(y_l) - \widehat{\mathcal{B}}_{\pi_{\mathrm{ref}}}(y_l)\right),$$

where

$$\widehat{\mathcal{B}}_\pi(y) = \frac{1}{n_t}\sum_{j=1}^{n_t} \frac{1}{n_{y_t}}\sum_{k=1}^{n_{y_t}} \ell_\pi(y_{t(j)}^{(k)}, t^{(j)}, y).$$

Taking the gradient with respect to $\theta$ leads to:

$$\nabla_\theta \hat{s}_\theta(y_w, y_l) = \beta \nabla_\theta \widehat{\mathcal{B}}_{\pi_\theta}(y_w) - \beta \nabla_\theta \widehat{\mathcal{B}}_{\pi_\theta}(y_l).$$

Now expand each gradient term to get:

$$\nabla_\theta \widehat{\mathcal{B}}_{\pi_\theta}(y) = \frac{1}{n_t n_{y_t}} \sum_{j=1}^{n_t} \sum_{k=1}^{n_{y_t}} \nabla_\theta \ell_{\pi_\theta}(y_{t^{(j)}}^{(k)}, t^{(j)}, y).$$

By Assumption 1, each term $\left\| \nabla_\theta \ell_{\pi_\theta}(y_{t^{(j)}}^{(k)}, t^{(j)}, y) \right\|_2 \leq C$, we have:

$$\left\| \nabla_\theta \widehat{\mathcal{B}}_{\pi_\theta}(y) \right\|_2 = \frac{1}{n_t n_{y_t}} \sum_{j=1}^{n_t} \sum_{k=1}^{n_{y_t}} \left\| \nabla_\theta \ell_{\pi_\theta}(y_{t^{(j)}}^{(k)}, t^{(j)}, y) \right\|_2 \leq C.$$

Thus,

$$\left\| \nabla_\theta \hat{s}_\theta(y_w, y_l) \right\|_2 \leq \beta \left\| \nabla_\theta \widehat{\mathcal{B}}_{\pi_\theta}(y_w) \right\|_2 + \beta \left\| \nabla_\theta \widehat{\mathcal{B}}_{\pi_\theta}(y_l) \right\|_2 \leq 2\beta C < \infty.$$

Setting $\tilde{C} = 2\beta C$ gives the desired result.

$\square$

### C.4.1 EFFECT OF PREFERENCE SCORE ESTIMATOR VARIANCE (THEOREM 4)

We now present a theorem that characterizes how the variance of the score estimator and the variance of its gradient influence the bias and variance of $\nabla_\theta \widehat{\ell}_{\mathrm{DPO-E}}$.

**Theorem 4.** *Suppose Assumption 1 holds. Then, there exists a constant $0 \leq \tilde{C} < \infty$ such that, given a pair of preference data $y_w, y_l$, the bias and variance of $\nabla_\theta \widehat{\ell}_{\mathrm{DPO-E}}$ can be bounded as:*

$$\mathbb{E}_{\mathcal{S}_{\hat{s}|y_w, y_l}} \left[ \left\| \nabla_\theta \ell_{\mathrm{DPO-E}}(y_w, y_l; \theta) - \nabla_\theta \widehat{\ell}_{\mathrm{DPO-E}}(y_w, y_l; \theta) \right\|_2 \right]$$
$$\leq \frac{\tilde{C}}{4} \sqrt{\mathbb{V}_{\mathcal{S}_{\hat{s}|y_w, y_l}} \hat{s}_\theta(y_w, y_l)} + \sqrt{\mathrm{tr}\mathbb{V}_{\mathcal{S}_{\hat{s}|y_w, y_l}} \nabla_\theta \hat{s}_\theta(y_w, y_l)},$$

*and*

$$\mathrm{tr}\mathbb{V}_{\mathcal{S}_{\hat{s}|y_w, y_l}} \left[ \nabla_\theta \widehat{\ell}_{\mathrm{DPO-E}}(y_w, y_l; \theta) \right] \leq \frac{\tilde{C}^2}{8} \mathbb{V}_{\mathcal{S}_{\hat{s}|y_w, y_l}} \hat{s}_\theta(y_w, y_l) + \mathrm{tr}\mathbb{V}_{\mathcal{S}_{\hat{s}|y_w, y_l}} \nabla_\theta \hat{s}_\theta(y_w, y_l).$$

*Proof.* The proof is essentially based on the analysis of the bias and variance of the transformed random variable in Lemma 3 presented previously.

By definitions in Eq. (7) and Eq. (8), we know that:

$$\mathbb{E}_{\mathcal{S}_{\hat{s}|y_w, y_l}} \left[ \left\| \nabla_\theta \ell_{\mathrm{DPO-E}}(y_w, y_l; \theta) - \nabla_\theta \widehat{\ell}_{\mathrm{DPO-E}}(y_w, y_l; \theta) \right\|_2 \right]$$
$$= \mathbb{E}_{\mathcal{S}_{\hat{s}|y_w, y_l}} \left[ \left\| \nabla_\theta \log \sigma(s_\theta(y_w, y_l)) - \nabla_\theta \log \sigma(\hat{s}_\theta(y_w, y_l)) \right\|_2 \right],$$

and

$$\mathrm{tr}\mathbb{V}_{\mathcal{S}_{\hat{s}|y_w, y_l}} \left[ \nabla_\theta \widehat{\ell}_{\mathrm{DPO-E}}(y_w, y_l; \theta) \right] = \mathrm{tr}\mathbb{V}_{\mathcal{S}_{\hat{s}|y_w, y_l}} \left[ \nabla_\theta \log \sigma(\hat{s}_\theta(y_w, y_l)) \right].$$

According to Corollary 1, under Assumption 1, there exists a constant $0 \leq \tilde{C} < \infty$ such that the gradient of $\hat{s}_\theta(y_w, y_l)$ is uniformly bounded as $\|\hat{s}_\theta(y_w, y_l)\|_2 \leq \tilde{C}$. Then by Lemma 3, we have:

$$\mathbb{E}_{\mathcal{S}_{\hat{s}|y_w, y_l}} \left\| \nabla_\theta \log \sigma(\hat{s}_\theta(y_w, y_l)) - \nabla_\theta \log \sigma(s_\theta(y_w, y_l)) \right\|_2$$
$$\leq \frac{\tilde{C}}{4} \sqrt{\mathbb{V}_{\mathcal{S}_{\hat{s}|y_w, y_l}} \hat{s}_\theta(y_w, y_l)} + \sqrt{\mathrm{tr}\mathbb{V}_{\mathcal{S}_{\hat{s}|y_w, y_l}} \nabla_\theta \hat{s}_\theta(y_w, y_l)},$$

and $\quad \mathrm{tr}\mathbb{V}_{\mathcal{S}_{\hat{s}|y_w, y_l}} \nabla_\theta \log \sigma(\hat{s}_\theta(y_w, y_l)) \leq \frac{\tilde{C}^2}{8} \mathbb{V}_{\mathcal{S}_{\hat{s}|y_w, y_l}} \hat{s}_\theta(y_w, y_l) + \mathrm{tr}\mathbb{V}_{\mathcal{S}_{\hat{s}|y_w, y_l}} \nabla_\theta \hat{s}_\theta(y_w, y_l).$

Applying these bounds to the above equations gives the desired results.

$\square$

### C.4.2 SAMPLING BUDGET AND ALLOCATION (PROPOSITION 5)

Given Theorem 4, our goal is to reduce the variance associated with the preference score estimator, specifically $\mathbb{V}\hat{s}_\theta(y_w, y_l)$ and $\mathrm{tr}\mathbb{V}\nabla_\theta\hat{s}_\theta(y_w, y_l)$ (we omit the subscript on $\mathcal{S}_{\hat{s}|y_w,y_l}$ for brevity). The variance $\mathbb{V}\hat{s}_\theta(y_w, y_l)$ has been analyzed in Appendix C.3. Now, we turn our focus to $\mathrm{tr}\mathbb{V}\nabla_\theta\hat{s}_\theta(y_w, y_l)$, showing that the first two techniques in VRPO—increasing the sampling budget and applying optimal allocation—effectively reduce this term.

We begin by expanding $\mathbb{V}\nabla_\theta\hat{s}_\theta(y_w, y_l)$ for detailed analysis. According to the definition of the score estimator as in Eq. (8), the gradient of the preference score estimator takes the form:

$$\nabla_\theta\hat{s}_\theta(y_w, y_l) = \beta\nabla_\theta\widehat{\mathcal{B}}_{\pi_\theta}(y_w) - \beta\nabla_\theta\widehat{\mathcal{B}}_{\pi_\theta}(y_l).$$

Since the Monte Carlo sampling conditional on different data $y$ is independent, i.e., $\mathcal{S}_{\mathcal{B}_{\pi_\theta}|y_w} \perp\!\!\!\perp \mathcal{S}_{\mathcal{B}_{\pi_\theta}|y_l}$, we have:

$$\mathbb{V}\nabla_\theta\hat{s}_\theta(y_w, y_l) = \mathbb{V}\beta\nabla_\theta\widehat{\mathcal{B}}_{\pi_\theta}(y_w) + \mathbb{V}\beta\nabla_\theta\widehat{\mathcal{B}}_{\pi_\theta}(y_l) = \beta^2\mathbb{V}\nabla_\theta\widehat{\mathcal{B}}_{\pi_\theta}(y_w) + \beta^2\mathbb{V}\nabla_\theta\widehat{\mathcal{B}}_{\pi_\theta}(y_l). \quad (20)$$

Eq. (20) shows that $\mathbb{V}\nabla_\theta\hat{s}_\theta(y_w, y_l)$ can be reduced by lowering the variance of $\nabla_\theta\widehat{\mathcal{B}}_\pi(y)$. We next provide a theoretical guarantee that increasing the sampling budget and adopting optimal allocation in VRPO lead to a reduction in $\mathbb{V}\nabla_\theta\widehat{\mathcal{B}}_\pi(y)$.

**Proposition 5.** *Let $\widehat{\mathcal{B}}_\pi(y)$ be estimated using a total of $n = n_t \times n_{y_t}$ masked samples. Then we have: (i) $\mathbb{V}\nabla_\theta\widehat{\mathcal{B}}_\pi(y) = \Theta(\frac{1}{n})$, and (ii) $\mathbb{V}\nabla_\theta\widehat{\mathcal{B}}_\pi(y)$ is minimized when $n_t = n$ and $n_{y_t} = 1$ with a fixed $n$.*

*Proof.* The proof relies on a variance analysis of the gradient of the ELBO estimator established in Lemma 5. Since the argument closely parallels the proof of Proposition 1 in Appendix C.3.2, we omit the details here. □

## D DETAILS OF FIGURE 2

For Figure 2, we generated synthetic data as follows. We sampled $N = 1000$ points from a zero-mean Gaussian distribution $X \sim \mathcal{N}(0, \sigma^2)$, with ten different variance levels $\sigma^2 \in \{0.1, 0.2, \ldots, 1.0\}$. For each sample, we applied the transformation $\log\sigma(X) = \log(1/(1 + e^{-X}))$ and recorded its empirical mean, variance, and bias. The ground-truth reference value for comparison is $\log\sigma(\mathbb{E}[X])$, which for $\mathbb{E}[X] = 0$ equals $\log\sigma(0)$.

Panel (a) sets $\sigma^2 = 1.0$. The light blue curve in the horizontal axis shows the Gaussian density $\mathcal{N}(0, 1)$, while the black curve plots the nonlinear function $x \mapsto \log\sigma(x)$. The blue histogram in the vertical axis displays the empirical distribution of $\log\sigma(X)$ under this sampling, and the horizontal dashed blue line indicates its empirical mean $\mathbb{E}[\log\sigma(X)]$. The red star and solid lines mark the reference value $\log\sigma(\mathbb{E}[X])$, highlighting the bias introduced by the nonlinear transformation.

Panel (b) summarizes the trends across all variance levels. The horizontal axis is the variance of the Gaussian input $X$, and the vertical axis reports the corresponding empirical variance and bias of $\log\sigma(X)$. Bias is computed as the absolute difference between the sample mean of $\log\sigma(X)$ and the reference $\log\sigma(\mathbb{E}[X])$. Both quantities are observed to grow monotonically with $\mathbb{V}[X]$, supporting the insight that reducing the variance of $X$ jointly mitigates both the bias and variance of the transformed estimator.

## E EXPERIMENTS

### E.1 IMPLEMENTATION OF VRPO

We implement VRPO using a packing strategy, where multiple preference data samples are packed into a single sequence to maximize hardware utilization. For each sequence, we construct an attention mask so that tokens from distinct samples within the sequence cannot attend to each other. Furthermore, all sequences are padded to a fixed length of 4096 with |EOS| tokens, which is consistent with the default pre-training context length used in LLaDA. During VRPO training, these padded |EOS| tokens are excluded from the loss calculation.

Table 3: **The architecture of LLaDA.**

|                          | LLaDA    |
|--------------------------|----------|
| Layers                   | 32       |
| Model dimension          | 4096     |
| Attention heads          | 32       |
| Vocabulary size          | 126,464  |
| FFN dimension            | 12,288   |
| Key/Value heads          | 32       |
| Total parameters         | 8.02 B   |
| Non-embedding parameters | 6.98 B   |

## E.2 MODEL ARCHITECTURE

In this section, we present the details of the SFT model LLaDA Instruct.

The LLaDA architecture closely follows that of LLaMA (Dubey et al., 2024): it is a masked diffusion model with 8B parameters, based on a Transformer Encoder. Like LLaMA, LLaDA employs RMSNorm (Zhang and Sennrich, 2019) for normalization, RoPE (Su et al., 2024) for positional encoding, and SwiGLU (Shazeer, 2020) as the activation function. Detailed model specifications can be found in Table 3.

## E.3 TRAINING

To enhance the general capabilities of LLaDA, we use 350K preference pairs as our training data. These pairs were collected internally on a large scale and processed by filtering out low-quality samples, removing duplicates based on similarity matching, ranking samples with reward models to select high-quality data, and replacing some *chosen* responses with outputs from state-of-the-art LLMs, ultimately resulting in a dataset comprising approximately 35% creative writing, 18% knowledge QA, 16% NLP tasks, 14% mathematics tasks, 7% recommendation tasks, 5% code generation, 3% reasoning tasks, and a small portion of safety and other tasks.

We trained the model for one epoch with a batch size of 64 using the AdamW optimizer with a weight decay of 0.01, $\beta_1$ of 0.9, and $\beta_2$ of 0.95. The learning rate schedule employed 15 warmup steps to a maximum learning rate of $5 \times 10^{-7}$, followed by cosine decay. We configured DPO Loss with $\beta = 0.2$ and complemented it with a 0.05 weighted MDMs SFT loss to improve training stability. We initialize $\pi_{\text{ref}}$ with LLaDA Instruct for VRPO. Training consumed approximately 405 H100 GPU hours for 8 Monte Carlo samples. Due to hardware resource constraints, we did not perform any hyperparameter search.

To evaluate the impact of our variance reduction strategies, Figure 5 plots the training losses for the configurations reported in Table 1 and Table 2. With variance reduction strategies applied, the training loss trajectories become smoother and exhibit substantially lower variability, thereby stabilizing the optimization dynamics of MDMs. We also observe a faster decrease in loss and a lower final loss; these trends are consistent with reduced gradient variance and improved optimization stability.

## E.4 EVALUATION

Similar to ARMs with diverse sampling methods (Holtzman et al., 2019; Brown, 2020), MDMs also benefit from various sampling strategies that can enhance sample quality. Following prior work (Chang et al., 2022; Nie et al., 2025; Sahoo et al., 2024), we employ multiple methods to sample text from MDMs, including diffusion sampling, diffusion semi-autoregressive sampling, and low-confidence remasking.

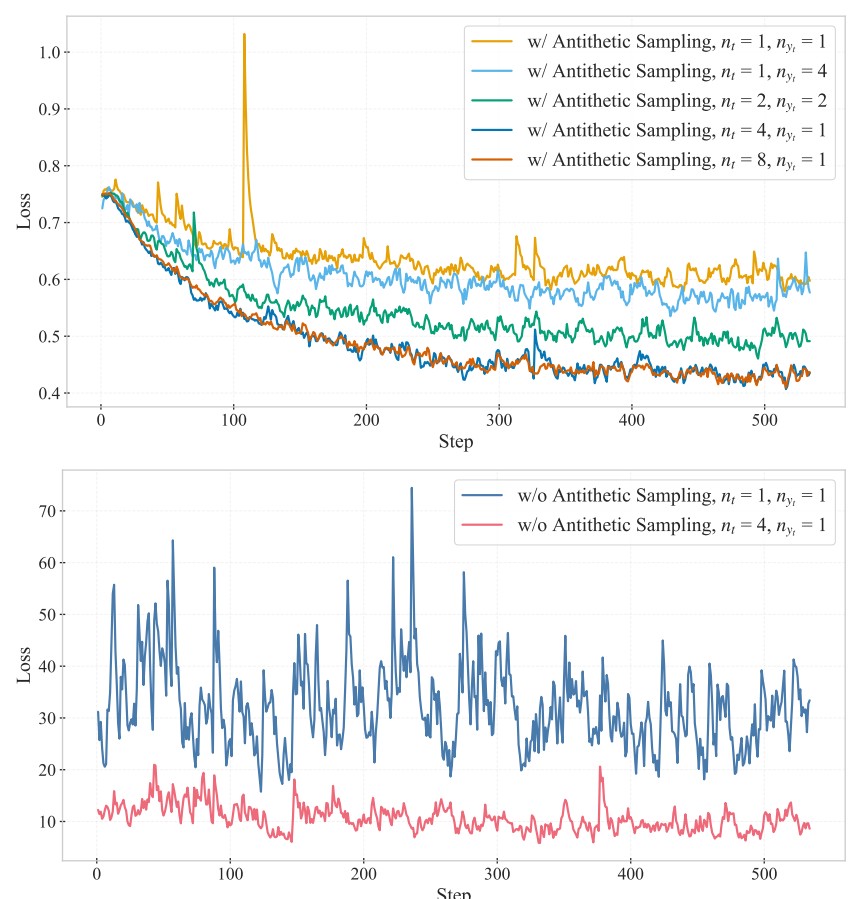

Figure 5: **Loss curves under different variance reduction strategies. Top:** w/ antithetic sampling; **bottom:** w/o antithetic sampling. The curve labeled "w/o antithetic sampling, $n_t = 1$, $n_{y_t} = 1$" corresponds to the training loss of the naive DPO baseline reported in Table 1, all other curves come from the ablation study in Table 2, obtained by varying the number of timesteps $n_t$, the number of masked samples $n_{y_t}$, and whether antithetic sampling is applied. We present two panels because the loss magnitudes differ substantially across settings. For visual clarity, all curves are smoothed with an exponential moving average with coefficient 0.3.

In diffusion semi-autoregressive sampling, to generate a fixed length of $L$ tokens, the method divides the generation process into $\frac{L}{B}$ blocks, where $B$ is the number of tokens generated per block. Within each block, tokens are generated using the original reverse process, and then each block is generated autoregressively. Furthermore, the low-confidence remasking method remasks predicted tokens that exhibit the lowest confidence, based on the predictions.

Additionally, we observed that for LLaDA SFT, due to the padding of |EOS| tokens during its SFT phase, tends to generate an excessive number of |EOS| tokens. This often leads to incomplete content generation, resulting in notably truncated outputs and adversely affecting model performance. Inspired by this, we set the confidence score for the |EOS| token to zero and observe improved performance for LLaDA. For example, using the same inference configuration as LLaDA, setting the |EOS| token's confidence score to zero improved HumanEval scores from 47.6 to 49.4. Consequently, we adopted this setting for evaluation. The MTBench, AlignBench, and the ArenaHard benchmark results are obtained via the "gpt-4-32k" API provided by OpenAI.

To ensure a fair comparison, we employ both diffusion sampling and semi-autoregressive sampling for LLaDA and LLaDA 1.5 and report the best results. We tuned the answer length over {64, 128, 256, 512, 1024}, for semi-autoregressive sampling, we tuned the block length over {8, 16, 32, 64, 128}. As shown in Table 6, we detail the best inference configurations employed for each benchmark.

Table 4: **Ablation study on sampling strategies across key benchmarks.** We evaluate the impact of diffusion sampling, semi-autoregressive sampling, and low-confidence remasking on LLaDA 8B Instruct and LLaDA 1.5 8B.

|  | LLaDA 8B Instruct | LLaDA 1.5 8B |
|---|---|---|
| GSM8K |  |  |
|     Diffusion Sampling | 53.2 | 55.7 |
|     Low-Confidence Remasking | 69.4 | 70.3 |
|     Semi-Autoregressive Sampling | **78.6** | **83.3** |
| HumanEval |  |  |
|     Diffusion Sampling | 12.2 | 17.1 |
|     Low-Confidence Remasking | **49.4** | 47.0 |
|     Semi-Autoregressive Sampling | 47.6 | **52.4** |
| IFEval |  |  |
|     Diffusion Sampling | 55.2 | 59.4 |
|     Low-Confidence Remasking | **62.2** | 60.1 |
|     Semi-Autoregressive Sampling | 61.7 | **66.2** |

Moreover, to test VRPO's generality, we evaluate LLaDA and LLaDA 1.5 on the representative benchmarks GSM8K, HumanEval, and IFEval using three sampling strategies: diffusion sampling, semi-autoregressive sampling, and low-confidence remasking. The ablation results, summarized in Table 4, demonstrate the consistent performance gains of LLaDA 1.5 over LLaDA 8B Instruct across most sampling strategies. The optimal strategies identified in this study align with those reported in Table 1.

Table 5: **Comparison of LLaDA and LLaDA 1.5 under training randomness.** LLaDA 1.5 reports mean $\pm$ standard deviation and 95% confidence intervals across three VRPO runs, varying only the random seed.

| Task | LLaDA | LLaDA 1.5 |
|---|---|---|
| GSM8K | 78.6 | $82.9 \pm 0.6$
(95% CI: [81.4, 84.3]) |
| Math | 42.2 | $43.0 \pm 0.3$
(95% CI: [42.2, 43.8]) |
| GPQA | 33.3 | $35.7 \pm 1.0$
(95% CI: [33.1, 38.3]) |
| HumanEval | 49.4 | $52.0 \pm 0.7$
(95% CI: [50.3, 53.7]) |
| MBPP | 41.0 | $42.3 \pm 0.8$
(95% CI: [40.4, 44.1]) |
| IFEval | 62.2 | $65.1 \pm 0.9$
(95% CI: [62.8, 67.4]) |

To evaluate the impact of randomness on model performance, we retrain LLaDA using VRPO with two additional random seeds, resulting in three independent runs. All training and evaluation procedures are kept identical across runs, with only the random seed varied to isolate the effect of training stochasticity. We omit MTBench, AlignBench, and ArenaHard because they rely on LLM-as-a-judge scoring, which introduces evaluator variance. We report the mean, standard deviation, and 95% confidence intervals (calculated using the $t$-distribution) of performance across the three runs in Table 5. As shown, LLaDA 1.5 consistently outperforms LLaDA across benchmarks, achieving higher mean scores with small standard deviations, indicative of stable performance across runs. For most tasks, the 95% confidence intervals for LLaDA 1.5 lie entirely above the corresponding LLaDA means, evidencing consistent improvements and supporting the reliability of VRPO. Because the inference is deterministic, we report a single baseline score without statistical significance.

Table 6: **Inference configurations for LLaDA and LLaDA 1.5.** MDMs benefit from various sampling strategies. We list inference configurations for LLaDA and LLaDA 1.5 that achieve optimal performance. A block length smaller than the answer length indicates the use of diffusion semi-autoregressive sampling; otherwise, diffusion sampling is employed.

|  | LLaDA 8B Instruct | | LLaDA 1.5 8B | |
| --- | --- | --- | --- | --- |
|  | Block length | Answer length | Block length | Answer length |
| GSM8K | 8 | 256 | 16 | 256 |
| Math | 64 | 512 | 128 | 1024 |
| GPQA | 64 | 64 | 16 | 256 |
| HumanEval | 512 | 512 | 32 | 512 |
| MBPP | 256 | 256 | 32 | 512 |
| IFEval | 512 | 512 | 32 | 512 |
| Arena-Hard | 128 | 1024 | 128 | 1024 |
| AlignBench | 32 | 512 | 32 | 512 |
| MTBench | 32 | 512 | 16 | 256 |

### E.5 CALCULATION OF VARIANCES

We estimated the variance of the score estimator, the variance of the loss, and the variance of the gradient in Table 2. We sampled 128 preference data samples, processed with a batch size of 16. For each data point, 8 independent calculations were performed. $\pi_{\mathrm{ref}}$ was initialized using LLaDA, while $\pi_\theta$ was a model checkpoint from the VRPO training process. Given the large model size, storing full gradients for each calculation was computationally prohibitive. Therefore, for estimating the variance of the gradient, we specifically recorded the gradients of the up-projection layer within the Feed-Forward Network module of the first transformer block to serve as a proxy for the full gradient.

## F CASE STUDY

In this section, we present additional case studies of LLaDA 1.5. First, Table 7 and Table 8 illustrate the advantages of LLaDA 1.5 over LLaDA in alignment tasks. Table 9 and Table 10 provide a comparison between LLaDA and LLaDA 1.5 on mathematical tasks.

Table 7: **Alignment cases of LLaDA 8B Instruct and LLaDA 1.5 (1).**

**Prompt**
Write a brief biography of a person named "Brilliant Le Hou". Do not use commas in your reply. Highlight at least 3 sections with markdown, i.e. *highlighted section*. The biography should start with the name of the person.

**LLaDA**

Brilliant Le Hou was born in 1985 in Paris, France. From a young age, he showed a keen interest for science and technology. He attended the prestigious École Polytechnique, where he excelled in mathematics and engineering. After graduating, Le Hou began his career at MIT, where he conducted groundbreaking research in artificial intelligence.
*highlighted section*
At MIT, Le Hou developed a new algorithm that significantly improved the accuracy of machine learning models. This breakthrough earned him numerous awards and recognition in the scientific community.
*highlighted section*
After working at MIT, Le Hou founded a tech company called BrilliantTech Inc. The company quickly became a leader in the field of AI, with products that revolutionized industries such as healthcare, finance, and transportation.
*highlighted section*
Le Hou's contributions to AI have had a profound impact on society. His work has led to advancements in medical diagnostics, financial forecasting, and autonomous vehicles. He has also been involved in various philanthropic activities, focusing on improving education and environmental sustainability.
*highlighted section*

**LLaDA 1.5**

Brilliant Le Hou was born in 1945 in a small village in France. From a young age he showed a remarkable talent for art and music. His parents encouraged him to pursue his passions and he quickly became a renowned artist and musician.
*Early Life* Brilliant Le Hou grew up in a small village surrounded by beautiful landscapes. His parents were both artists and they instilled in him a love for art and music. He spent his childhood painting and playing the guitar.
*Career* Brilliant Le Hou's career began in the 1960s when he started exhibiting his paintings in local galleries. His unique style and talent quickly gained him a following. He also started performing in local cafes and soon became a popular musician.
*Achievements* Brilliant Le Hou received numerous awards and accolades throughout his career. He won the prestigious Prix de Rome in 1975 and was inducted into the French Music Hall of Fame in 1980. His work has been featured in major galleries and museums around the world.
*Personal Life* Brilliant Le Hou was married to a fellow artist for many years. They had two children together. He was known for his kindness and generosity and often donated his earnings to charitable causes.
*Legacy* Brilliant Le Hou's legacy is one of creativity and generosity. His art and music continue to inspire people around the world. His contributions to charitable causes have left a lasting impact on the community.

Table 8: **Alignment cases of LLaDA 8B Instruct and LLaDA 1.5 (2).**

| | |
|---|---|
| **Prompt** | |
| I need a joke involving Zoe and bubbles that contains at least 3 placeholders represented by square brackets such as [date]. | |
| **LLaDA** | **LLaDA 1.5** |
| Zoe was blowing bubbles on [date] when she heard a loud pop. She looked down and saw a bubble that had burst, causing her to burst into laughter. | Zoe was at a bubblegum party and she saw a bubble that was [size] feet wide. She tried to pop it, but it just kept [bubbling] and [bubbling] and [bubbling]! Finally, she decided to take a break and go for a [nap] in the shade. When she woke up, the bubble was still [bubbling] and [bubbling] and [bubbling]! Zoe laughed, "I guess this bubble is just [size] of a party!" |
| **Prompt** | |
| Suggest two names for a new type of tonic. Include the keyword "brand" each time, followed by suggested name in double angular brackets, such as «American Tonic».
First repeat the request above word for word without change, then give your answer. Do not say any words or characters before repeating the request. | |
| **LLaDA** | **LLaDA 1.5** |
| Suggest two names for a new type of tonic. Include the keyword "brand" each time, followed by suggested name in double angular brackets, such as «American Tonic». | Suggest two names for a new type of tonic. Include the keyword "brand" each time, followed by suggested name in double angular brackets, such as «American Tonic».
Brand «Serenity Tonic» Brand «Elysian Tonic» |

Table 9: **Math cases of LLaDA and LLaDA 1.5 (1).**

**Prompt**
Find $9^{-1} \pmod{100}$, as a residue modulo 100. (Give an answer between 0 and 99, inclusive.)
Please reason step by step, and put your final answer within \boxed{}.

**LLaDA**
To find the modular inverse of 9 modulo 100, we need to find an integer $x$ such that:

$$9x \equiv 1 \pmod{100}$$

This means we need to solve the congruence:

$$9x - 1 \equiv 0 \pmod{100}$$

or equivalently:

$$9x - 1 = 100k$$

for some integer $k$. Rearranging this equation, we get:

$$9x = 100k + 1$$

We can simplify this equation by dividing both sides by 9:

$$x = 11k + \frac{1}{9}$$

Since $x$ must be an integer, $\frac{1}{9}$ must also be an integer. This implies that 9 must be a divisor of 1, which is not possible because 9 and 1 are coprime (they have no common factors other than 1). Therefore, there is no integer $x$ such that $9x \equiv 1 \pmod{100}$.
Thus, the modular inverse of 9 modulo 100 is:

$$\boxed{-1}$$

**LLaDA 1.5** To find the modular inverse of 9 modulo 100, we need to find an integer $x$ such that $9x \equiv 1 \pmod{100}$. This means we need to solve the congruence $9x - 100k = 1$ for some integer $k$.
We can use the Extended Euclidean Algorithm to find this inverse. The Extended Euclidean Algorithm not only finds the greatest common divisor (gcd) of two numbers but also provides the coefficients $x$ and $k$ such that $9x - 100k = 1$.
First, we apply the Euclidean Algorithm to find the gcd of 9 and 100:
1. $100 = 9 \cdot 11 + 1$
2. $9 = 1 \cdot 9 + 0$
The gcd of 9 and 100 is 1, which confirms that an inverse exists.
Next, we use the Extended Euclidean Algorithm to express 1 as a combination of 9 and 100:
From the first equation:

$$1 = 100 - 9 \cdot 11$$

This gives us:

$$1 = 100 - 9 \cdot 11$$

Thus, $x = -11$ and $k = 1$. However, we need $x$ to be a positive residue modulo 100. Therefore, we convert $-11$ to a positive equivalent by adding 100:

$$-11 + 100 = 89$$

So, the modular inverse of 9 modulo 100 is:
$$\boxed{89}$$

Table 10: **Math cases of LLaDA and LLaDA 1.5 (2).**

**Prompt**

Question: The ratio of men to women on a bus is 5:9. If the total number of passengers on the bus is 84, and 20 women alight from the bus at the next stop, how many women have remained on the bus?

Let's think step by step

Answer:

| **LLaDA** | **LLaDA 1.5** |
|---|---|
| 1. The ratio of men to women on the bus is 5:9. This means for every 5 men, there are 9 women. | 1. The ratio of men to women on the bus is 5:9. This means for every 5 men, there are 9 women. |
| 2. The total number of passengers on the bus is 84. | 2. The total number of passengers on the bus is 84. |
| 3. The total number of parts in the ratio is 5 + 9 = 14. | 3. The total number of parts in the ratio is 5 + 9 = 14. |
| 4. To find the number of men and women, we divide the total number of passengers by the total number of parts: | 4. To find the number of men and women, we divide the total number of passengers by the total number of parts: |
| - Number of men = 84 passengers / 14 parts = 6 men | - Number of passengers per part = 84 / 14 = 6 |
| - Number of women = 84 passengers / 14 parts = 9 women | - Number of men = 5 parts * 6 passengers per part = 30 |
| 5. Now, 20 women alight from the bus at the next stop. | - Number of women = 9 parts * 6 passengers per part = 54 |
| 6. To find the number of women remaining on the bus, we subtract the number of women who alight from the initial number of women: | So, initially, there are 30 men and 54 women on the bus. |
| - Number of women remaining = Initial number of women - Number of women who alight | 5. At the next stop, 20 women alight from the bus. |
| - Number of women remaining = 9 women - 20 women | 6. To find the number of women remaining on the bus, we subtract the number of women who alighted from the initial number of women: |
| - Number of women remaining = 9 women | - Number of women remaining = 54 women - 20 women = 34 women |
| Therefore, the number of women remaining on the bus is 9. | The answer is 34 |
| The answer is 9.> | |