# OpenReview forum: "LLaDA 1.5: Variance-Reduced Preference Optimization for Large Language Diffusion Models"
_ICLR.cc/2026/Conference — ICLR 2026 Conference Withdrawn Submission_

### Official Review · Reviewer_gNKy · 2025-10-28

**Soundness:** 3
**Presentation:** 3
**Contribution:** 1
**Rating:** 2
**Confidence:** 3

**Summary:**

This paper presents VRPO (Variance-Reduced Preference Optimization) and applies it to produce LLaDA 1.5, a preference-optimized masked diffusion language model. VRPO combines three variance-reduction strategies for ELBO-based preference estimation: (1) increasing the Monte Carlo sampling budget for ELBOs, (2) allocating that budget across diffusion timesteps (nt/nyt scheduling), and (3) applying antithetic (paired) sampling between model and reference ELBO estimates. The paper provides variance decomposition analysis relating score-estimator variance to loss/gradient variance, give propositions showing unbiased variance reduction from these strategies. Empirically, the paper shows LLaDA 1.5 achieves consistent improvements over both SFT-only and naïve DPO baselines across various tasks

**Strengths:**

- The paper points out that ELBO-based DPO alignment introduces bias and variance coupling that degrades optimization stability.
- Section 4.2 shows ablation study by each three components
- Empirical results support the claim that the proposed adjustments improve training dynamics.

**Weaknesses:**

- The method is largely a combination of well-known variance reduction techniques, meaning methodological novelty is limited.
- There are computation overhead from sampling increase. The paper admits this overhead yet does not convincingly show superiority under equal resource conditions
- Ablation study is limited. Individual contributions of each component are not clearly quantified.

**Questions:**

- The performance of VRPO when applied to other model (e.g. GRPO?)?
- each nt/nyt hyperparameter analysis?

---

### Official Review · Reviewer_hLJC · 2025-10-31

**Soundness:** 3
**Presentation:** 3
**Contribution:** 2
**Rating:** 6
**Confidence:** 3

**Summary:**

The paper "LLaDA 1.5: Variance-Reduced Preference Optimization for Large Language Diffusion Models" presents a new preference optimization framework called VRPO to enhance the alignment performance of diffusion language models (MDMs).

**Strengths:**

The paper has a complete structure and is generally well-written.

**Weaknesses:**

1.	The paper’s primary contribution lies in theoretically identifying the high-variance issue within the ELBO estimation as a key factor causing DPO’s instability in masked diffusion models (MDMs), and in proposing the VRPO framework to mitigate this variance. However, the main innovation resides in the problem formalization and attribution analysis rather than in the algorithmic design itself. The proposed variance-reduction techniques, while theoretically sound, are based on established statistical optimization principles and are applied in a relatively shallow manner.
Moreover, the paper focuses exclusively on variance minimization without discussing the bias–variance trade-off, which plays a crucial role in deep learning optimization. Allowing for a small amount of bias can sometimes improve convergence and generalization, and a deeper exploration of this trade-off would further strengthen the theoretical completeness of the work.
2.	Suggest that authors, when discussing the reasons for DPO's failure on MDM, broaden the analytical perspectives further. Currently, the paper mainly attributes it to "high variance" and is supported by detailed mathematical derivations; this analytical path is clear and insightful. However, the phenomenon may also involve more complex factors, such as semantic alignment issues between human-labeled data and the model's denoising training process, or the matching between the diffusion process's dynamics at different time steps and the objective of preference learning. If these aspects can be supplemented with discussion, it would help to more comprehensively understand the essence of the problem and provide a stronger theoretical foundation for future improvements.
3.	Experiments only compared "naive DPO" and lacked side-by-side comparisons with other alignment methods. It is recommended to add at least one comparative experiment with an improved DPO method, such as TDPO [1].
4.	The experimental results lack data on variance or significance tests, which are recommended to be added to demonstrate the robustness of the experimental effects.
5.	It is unclear whether the sampling budget scale affects computational efficiency and performance. It is suggested to include scaling analysis to verify the computational efficiency trade-offs of VRPO.
6.	In Section 4 "Data," the authors mention "internally collected 350K preference pairs" but do not disclose their distribution, source, or sampling rules. It is recommended to provide clearer data sources in the public version.

## Reference
[1] Zeng, Y., Liu, G., Ma, W., Yang, N., Zhang, H., & Wang, J. (2024). Token-level direct preference optimization (arXiv:2404.11999 [cs.CL]). arXiv. https://doi.org/10.48550/arXiv.2404.11999

**Questions:**

Please refer to the “Weakness” section for related questions.

---

### Official Review · Reviewer_6tgG · 2025-10-31

**Soundness:** 3
**Presentation:** 2
**Contribution:** 3
**Rating:** 4
**Confidence:** 4

**Summary:**

This paper studies a preference optimization technique for diffusion LLM. The challenge is that the likelihood is not readily available for diffusion models. They propose to plug in the ELBO as a proxy of likelihoods to the DPO loss. Since the overall objective is in the form of $\mathbb{E}\_Y[h(\mathbb{E}\_X[f(X,Y)|Y]]]$, where the expectations are approximated by Monte Carlo expectations, the empirical estimator suffers from bias and variance. The paper focuses on reducing the variance in the ELBO, to minimize the overall estimation error.
The paper proposes Variance-Reduced Preference Optimization (VRPO), which is a combination of "techniques" to minimize the variance.
The paper demonstrates the effectiveness of the proposal with ablation studies.

**Strengths:**

This paper studies a timely topic on how to align diffusion LLMs with preferential data.
While the writing is not particularly polished, the paper remains understandable overall.
The experiment is well designed show the efficacy of the proposal.

**Weaknesses:**

The proposed ideas in VRPO (i.e., optimal allocation + antithetic sampling) are not too surprising. (I think that the first item on sampling budget to increase the number of samples $n$ is too trivial and obvious to be credited to authors.) But since the authors theoretically and empirically demonstrate the effects of the techniques, this incremental contribution is somewhat justifiable.

Beyond the novelty, I have a few concerns on the framing and structure of the paper.
- The foremost one is how the authors frame the contributions. I find the title a bit misleading, as they put LLaDA 1.5 as their main contribution. I think their contribution is the generic variance reduction technique and analysis. LLaDA 1.5 seems to be a simple outcome out of their techniques on top of LLaDA. Also, if there is no prior work that tries to apply DPO for diffusion LLM, then that should have been clarified as the authors' core contribution.
- The terminology VRPO is also a bit misleading, as it's only specific for diffusion LLMs. But I think this is a rather minor point.
- The structure and flow of Section 3.2 are disorganized and make the argument difficult to follow. The main techniques (1), (2), (3) on page 5 should follow after the propositions. Or, the authors should have at least mentioned how they come up with such techniques.
- Proposition 1 (especially the statement (i)) is misleading. In (i), $n_t$ can be an arbitrary natural number that divides $n$. Then it should be stated that the variance is in the order of $\frac{1}{n\_t}$, which only makes logical sense to conclude that we should set $n\_t=n$ in (ii). This misleading statement stems from the proof of Proposition 1, where they introduce $c=n\_t/n$. I suggest the authors to revise the statement and the proof.

**Questions:**

- Can you remark the variational gap in ELBO? When does it become tight?
- In this paper, ELBO is plugged in place of the likelihood, playing as its direct proxy. However, considering that it's only a lower bound, directly plugging it into the DPO loss in Eq. (3) to get an estimator sounds only like a heuristic. Can the authors make a remark about this point?

#### **Suggestions**
- The use of em dashes in the last sentence of the second paragraph on page 1 is weird.
- The sentence `Computing B_π (y) exactly is generally intractable due to the double expectations.` is a bit misleading. I believe that the integral remains intractable even if you only have an integral over $y$. Double expectation only makes it harder.

---

I think the paper has practically useful ideas with nice analyses, though not rocket science. I believe that the manuscript can significantly benefit from careful revision with restructuring and reframing.

---

### Note · Authors · 2025-11-21

**Comment:**

We have decided to withdraw this submission for further improvement.

**Withdrawal Confirmation:**

I have read and agree with the venue's withdrawal policy on behalf of myself and my co-authors.